# Diagnosing the controls on desert dust emissions through the Phanerozoic

Yixuan Xie[1], Daniel J. Lunt[1], and Paul J. Valdes[1]

[1]School of Geographical Sciences, University of Bristol, Bristol BS8 1SS, UK

**Correspondence:** Yixuan Xie (yixuan.xie@bristol.ac.uk)

**Abstract.** Desert dust is a key component of the climate system, as it influences Earth's radiative balance and biogeochemical cycles. It is also influenced by multiple aspects of the climate system, such as surface winds, vegetation cover, and surface moisture. As such, geological records of dust deposition or dust sources are important paleoclimate indicators; for example, dust records can be used to decipher aridity changes over time. However, there are no comprehensive records of global dust variations on tectonic time scales (10's of millions of years). Furthermore, although some modelling studies have focused on particular time periods of Earth's history, there has also been very little modelling work on these long timescales. In this study, we establish for the first time a continuous model-derived timeseries of global dust emissions over the whole Phanerozoic (the last 540 million years). We develop and tune a new offline dust emission model, DUSTY1.0, driven by the climate model HadCM3L. Our results quantitatively reveal substantial fluctuations in dust emissions over the Phanerozoic, with high emissions in the Late Permian to Early Jurassic ($\times 4$ pre-industrial levels), and low emissions in the Devonian-Carboniferous ($\times 0.1$ pre-industrial levels). We diagnose the relative contributions from the various factors driving dust emissions and identify that the non-vegetated area plays a dominant role in dust emissions. The mechanisms of paleo hydrological variations, specifically the variations in low-precipitation-induced aridity, which primarily control the non-vegetated area, are then diagnosed. Our results show that paleogeography is the ultimate dominating forcing with the dust emissions variations explained by indices reflecting the land-to-sea distance of tropical and subtropical latitudes, whereas $CO_2$ plays a marginal role. We evaluate our simulations by comparing them with sediment records and find reasonable agreement. This study contributes a quantified and continuous dust emission reconstruction, as well as an understanding of the mechanisms driving paleohydroclimate and dust changes over Earth's Phanerozoic history.

## 1 Introduction

Dust plays a pivotal role in the Earth system. Dust affects the climate system in various ways. Dust modulates the Earth's radiation budget directly by scattering, absorbing and re-emitting radiation, but also indirectly regulates the Earth's radiation balance by stimulating cloud and precipitation formation processes (e.g. Schepanski, 2018). In addition, the deposition of dust onto the Earth's surface provides nutrients such as nitrogen, sulfur, iron and phosphorus, which affect the corresponding ecosystems and, ultimately, the global carbon cycle (Mahowald et al., 2017). For example, dust is the dominant external source

of iron to the ocean, and the supply of iron is a limiting factor on marine life in large parts of the ocean and therefore influences the ability of the ocean to regulate atmospheric $CO_2$ (Jickells et al., 2005).

On the other hand, the climate system regulates dust processes in various ways. Rainfall, evapotranspiration, vegetation and the associated vegetation cover have an effect on sediment availability (Marx et al., 2018). Soil moisture, sediment particle characteristics, snow cover and surface wind jointly control the dust entrainment (Kok et al., 2012). In addition, shifts in wind

regimes, such as the position of synoptic-scale circulation systems, and changes in clouds and precipitation at smaller-scale zones influence the transportation and deposition of dust (Marx et al., 2018).

The estimated global annual dust emission flux is about 2000 Mt, originating from the world's major deserts located in Africa, China and Mongolia, Australia, central and southwest Asia and the United States (Shao, 2001). The majority of the contemporary dust source regions are located in subtropical regions, which are associated with the position of the Hadley

circulation. Less rainfall is produced in extratropical latitudes Hadley Cell descending branch than its tropical ascending branch as a result of the meridional heat and energy transportation (Diaz and Bradley, 2004; Nguyen et al., 2013), and hence leads to the broadly distributed subtropical aridity. In addition, the contribution of dust emissions from high-latitude paraglacial dust source regions is well constrained, accounting for about 5% of the overall dust budget (Bullard et al., 2016).

Because arid areas are in general associated with relatively high dust emissions, geological records of changes in dust

emissions in Earth's past serve as a reliable archive for environmental and climate variations in aridity. In addition, past climates provide a natural laboratory for modelling studies which aim to explore controls on dust emissions.

Existing records of past dust emission/deposition show high variability through time. Reconstructions are most abundant in the Quaternary (the last 2.6 million years) and many of them indicate substantial glacial-interglacial variability – the Earth becomes dustier during glacial periods than interglacial periods (e.g. Lamy et al., 2014; Fitzsimmons et al., 2013) and records

compiled in Muhs (2013) and Kohfeld and Harrison (2001). Studies have indicated that the more substantial glacial dust deposits could be attributed to increased wind intensities, a less vigorous hydrological cycle, decreased soil moisture, decreased vegetation cover, and exposed continental shelves (Winckler et al., 2008; Muhs, 2013), all of which are ultimately caused by variations in the Earth's orbital parameters. Some studies highlight the importance of glaciogenic dust as a significant sediment source due to glacial grinding (e.g. Mahowald et al., 2006; Sugden et al., 2009), which contributes to the formation

of widespread loess deposits in the Quaternary (Li et al., 2020).

Similar orbital-driven dust (or aridity) records are also identified much further back in time, such as during the Late Cretaceous (Niedermeyer et al., 2010; Vallé et al., 2017; Zhang et al., 2019) and at the late Paleozoic (Soreghan et al., 2008). Moreover, the variability in dust (or aridity) on timescales older than the Quaternary is explained as responses to regional or global cooling (DeCelles et al., 2007; Bosboom et al., 2014; Licht et al., 2016; Zhang et al., 2016), atmospheric $CO_2$ concen-

tration variations (Wang et al., 2023), tectonic movements (Rea et al., 1998; Licht et al., 2016; Farnsworth et al., 2019; Zhu et al., 2020; Anderson et al., 2020; Yang et al., 2021; Lin et al., 2024), the absence of land vegetation (Liu et al., 2020), and intensified glaciation (McGlannan et al., 2022; Gabbott et al., 2010). However, these existing records and modelling studies have a relatively small scope, either spatially (only for a specific region) or temporally (only for a specific time period or a couple of time slices) or both, and there is also a lack of geological records of global dust variations through time. As such,

our knowledge of the dust (or aridity) distributions, variations and the underlying driving forces in the Earth's deep time still remain insufficiently understood.

The aim of this study is, therefore, to investigate (1) How do the global dust emissions change through the Phanerozoic? (2) What are the causes and mechanisms of these changes? In order to do this, we develop a new offline dust emission model – DUSTY1.0, which is then driven by the output from a set of 109 General Circulation Model (GCM) simulations covering the whole Phanerozoic (last 540 million years).

The DUSTY1.0 model and the underlying GCM and GCM simulations are described in section 2. The DUSTY1.0 model is tuned to improve its simulations of the modern, described in Section 3. The paleo dust emissions simulations covering the whole Phanerozoic are then carried out with the tuned DUSTY1.0 model. Results are shown in section 4, plus analyses of the contributing factors and the driving force of the Phanerozoic dust emissions variations. The credibility of our simulated dust emissions is evaluated with a comparison to geological records, with discussions of the implications and limitations of this study in section 5.

## 2    Methods

The following section describes the dust emission model "DUSTY1.0" developed for this study, the GCM that is used to drive the DUSTY1.0 model, and the simulations that have been carried out with the GCM.

### 2.1    The dust emission model "DUSTY1.0"

The DUSTY1.0 model developed for this study is an offline model. It is designed to be driven by dust-relevant meteorological variables at a relatively high temporal frequency (for our study, every 1 hour), either the output from a GCM or observations, in order to calculate dust emissions for a particular climate scenario. Applying an offline dust model can make the dust simulation more efficient than using a coupled dust scheme within a GCM - allowing multiple sensitivity studies to be carried out without having to run the GCM multiple times. Also, for this study, due to the fact that we are simulating multiple time periods, we focus only on dust emissions rather than transport and deposition, in order to save computational cost. The dust emission model is designed to match the same spatial and temporal resolution of its driving input.

The following dust-relevant fields are used to drive the DUSTY1.0 model: the land-sea distribution ($l$; equal to 0 or 1), the bare soil fraction of the land surface ($b$; varying from 0 to 1, non-vegetated area hereafter), the soil moisture ($m$; units ($kg\,m^{-2}$)), the snow cover ($s$; units ($kg\,m^{-2}$)) and the near-surface (10-meter height) wind velocity ($u$ and $v$; zonal and meridional respectively, both units ($m\,s^{-1}$)). The spatial resolution of the dust model for this study is 96 × 73 grid points, which is 3.75º × 2.5º in longitude and latitude respectively, corresponding to the driving climate model (see Section 2.2).

The choice of the variables mentioned above is similar to other dust models. Vegetation and snow cover are commonly considered to inhibit dust emission from land surfaces and are generally represented in dust models with empirical globally uniform threshold values or as a linear function with no threshold (Ginoux et al., 2001; Miller et al., 2006; Kok et al., 2014). The formulation in our dust emission model assigns a linear function and a threshold for both ($b_t$ and $s_t$ respectively), where

the value is defined by further tuning process (Equations 3 and 5 below). Soil moisture is widely considered to suppress dust entrainment. Here we represent this with a simplified threshold-based ($m_t$) linear formulation (Equation 4 below), instead of considering it as a factor of threshold friction velocity of soil particles as in previous dust schemes (Pérez et al., 2011; Ginoux et al., 2001). As shown by many studies (e.g., Bagnold, 1941; Gillette and Stockton, 1989; Shao et al., 1993), dust emissions can be expressed in terms of surface wind speed above a threshold ($U_{t1}$), and the dust emission flux is approximately proportional to the third power of the wind speed. Our dust emission model adapts the formulation developed by Gillette and Passi (1988), which is also widely used in dust models such as Tegen and Fung (1994); Ginoux et al. (2001); Kerkweg et al. (2006); Miller et al. (2006); Chen et al. (2017). We also add a maximum threshold ($U_{t2}$) for dust emissions to avoid spuriously large emissions when the surface wind speed is extremely strong (Equation 6). There are some factors omitted in this dust emission model compared to some existing dust schemes – the effect of particle size, their soil characteristics and mineralogical features that are believed to be factors for friction velocity of dust entrainment are not included in our dust emission model due to the absence of information on these aspects on the timescales we are addressing here. In addition, "preferential sources areas" (Zender et al., 2003; Cakmur et al., 2006) which are constrained by the surface topography as "hydrological basins", and the sediment availability related to the glacial grinding process (Prospero et al., 2002; Mahowald et al., 2006), are not considered in this dust emission model for the same reasons. As such, our model is supposed only to be able to test the hypothesis of dust emissions related to hydroclimate processes rather than the sediment supply hypothesis.

The total dust emissions index, $d$ (no units) of each grid box, is the product of each individually ascribed emission index (Equation 1). $d_l, d_b, d_m, d_s, d_U$ are corresponding emission indices associated solely with the land-sea distribution, the non-vegetated area, the soil moisture, the snow and the near-surface wind, as calculated by Equations 2 to 6 respectively.

$$d = d_l * d_b * d_m * d_s * d_U \tag{1}$$

$$d_l = l \tag{2}$$

$$d_b = \begin{cases} b, & \text{if } b > b_t \\ 0, & \text{otherwise} \end{cases} \tag{3}$$

$$d_m = \begin{cases} (m_t - m)/m_t, & \text{if } m < m_t \\ 0, & \text{otherwise} \end{cases} \tag{4}$$

$$d_s = \begin{cases} (s_t - s)/s_t, & \text{if } s < s_t \\ 0, & \text{otherwise} \end{cases} \tag{5}$$

$$U = \sqrt{u^2 + v^2}, \qquad\qquad d_U = \begin{cases} 0, & \text{if } U < U_{t1} \\ (U - U_{t1}) * U^2, & \text{if } U_{t1} \leq U \leq U_{t2} \\ (U_{t2} - U_{t1}) * U_{t2}^2, & \text{if } U > U_{t2} \end{cases} \qquad (6)$$

Specifically, the land-sea distribution gives the basis for subsequent calculations as all the dust emissions are only considered over land areas. The calculation of consequent dust emissions related to bare soil, soil moisture and snow cover are all threshold-based linear relational expressions. The corresponding index for dust emissions induced by surface winds is calculated as a 120 cubic relationship with two thresholds.

$$D = C_2 * d \qquad (7)$$

A coefficient $C_2$ is here used to calibrate the total dust emission for the specific grid box, $D$ (Equation 7), and to convert to units of kg m$^2$ s$^{-1}$. A globally averaged emission rate of $1.96 * 10^{-10}$ $kg\ m^{-2}s^{-1}$ is used (see details in section 3.1) to derive the coefficient $C_2$. This coefficient and the following coefficients introduced in the next paragraph are all calculated at a global 125 mean level.

In addition, The dust emission model also produces dust emissions with various combinations of those dust-relevant fields, either included or not included. Using the same naming convention as in equation 7, the emissions calculated are $D^l$, $D^{lb}$, $D^{lbm}$, $D^{lbms}$, $D^{lbU}$, $D^{lbmU}$, $D^{lbsU}$, as given in equation 8. Similar to $C_2$, different coefficients $(C_1, C_3, C_4, C_5)$ are used in order to scale the emissions with various process combinations. $C_1$ is chosen such that the global dust emissions in $D^{lbms}$ are 130 equal to $D^{lbmsU}$, $C_3, C_4, C_5$ are chosen such that the global dust emissions in $D^{lbU}$, $D^{lbmU}$ and $D^{lbsU}$ are equal to $D^{lbms}$.

$$\begin{aligned} D^l &= C_1 * d_l \\ D^{lb} &= C_1 * d_l * d_b \\ D^{lbm} &= C_1 * d_l * d_b * d_m \\ D^{lbms} &= C_1 * d_l * d_b * d_m * d_s \\ D^{lbU} &= C_3 * d_l * d_b * d_U \\ D^{lbmU} &= C_4 * d_l * d_b * d_m * d_U \\ D^{lbsU} &= C_5 * d_l * d_b * d_s * d_U \end{aligned} \qquad (8)$$

## 2.2 The GCM

For this study, the DUSTY1.0 model is driven by simulations carried out with a General Circulation Model (GCM) HadCM3BL-MOSES2.1a-TRIFFID (HadCM3BL-M2.1aD in the naming convention of Valdes et al. (2017); henceforth HadCM3L). This is 135 a coupled atmosphere-ocean-vegetation model. The atmospheric component of HadCM3L solves the primitive equation set of

White and Bromley (1995) through the Arakawa staggered B-grid scheme from Arakawa and Lamb (1977). Parameterisations include the convection scheme of Gregory et al. (1997), the large-scale precipitation scheme of Wilson (1998), the radiation scheme of Edwards and Slingo (1996), and the clouds scheme of Bushell (1998). The ocean component is based on the model of Cox et al. (1984), which is a three-dimensional ocean model. Parameterisations include the ocean mixed layer schemes of Turner and Kraus (1967). Modifications to the ocean vertical diffusion and isopycnal diffusion due to the model's lower resolution are described in detail in Valdes et al. (2017). There are also modifications in the bathymetry of the North Atlantic, which improves the reality of heat transports in the coupled system and alleviates the need for flux correction and makes the model more appropriate for paleo simulations.

The land surface scheme MOSES2.1 calculates the energy and moisture between the land surface and the atmosphere and updates the relevant surface and subsurface variables (Cox et al., 1999). It is coupled with an interactive vegetation model, TRIFFID, via nine land cover types: broadleaf trees, needleleaf trees, shrubs, C3 grasses, C4 grasses, urban, inland water, bare soil and land ice. TRIFFID takes the averaged flux of carbon from MOSES2.1, calculates the growth and expansion of five defined plant functional types and updates the vegetation fractions and parameters through competitive, hierarchical formulation. We do not change this representation of vegetation, even though grasses did not become dominant until after the end-Cretaceous, and C4 vegetation did not become dominant until much later. Our assumption is that other forms of ground cover vegetation were present in these older time periods and impacted climate in similar ways.

HadCM3L has a horizontal resolution of 96 × 73 grid points in both the atmosphere and the ocean, which is 3.75º × 2.5º in longitude and latitude, and 19 hybrid vertical levels in the atmosphere and 20 vertical levels in the ocean with finer definition closer to the surface. It is of relatively lower resolution compared to recent state-of-the-art CMIP6 models, therefore, it is particularly computationally efficient and applicable for multi-million-year scale simulations. The model has been used widely in pre-Quaternary climate modelling studies (e.g. Marzocchi et al., 2015; Wade et al., 2019; Jones et al., 2022).

Since Valdes et al. (2017), the model has undergone several improvements, the most significant of these is a tuning process designed in particular to improve the deep-time climate simulations. The changes made follow those described in Sagoo et al. (2013); Kiehl and Shields (2013), primarily targetting parameters associated with clouds; this results in a flatter meridional temperature gradient in warm climates, which is in better agreement with paleoclimate proxies. In addition, we have added a new representation of soil albedo. In the original model configuration, soil was specified as a medium loam everywhere. We have now added a soil carbon dependence. If the soil carbon content drops below 15 $kg\,m^{-2}$, the albedo linearly increases to a maximum of 0.355. These numbers were tuned so that the model could simulate the present-day average Saharan soil albedo well.

## 2.3   The GCM simulations

This study uses 3 series of 109 experiments each, $S_1$, $S_{1noCO2}$ and $S_2$, all carried out with HadCM3L, corresponding to roughly 109 geological stages covering the whole Phanerozoic (since 541 million years). Similar simulations are described in Valdes et al. (2021). However, the Valdes et al. (2021) simulations do not include the tuning processes described in section 2.2. Series $S_2$ does include those tuning, but is otherwise identical to series $S_1$. The boundary conditions to drive the $S_1$ and

$S_2$ simulations include (1) solar constant following Gough (1981); (2) atmospheric $CO_2$ concentrations following Foster et al. (2017); and (3) paleogeography (the configuration and height/depth of the continents and oceans) following Scotese and Wright (2018). The boundary conditions used to drive $S_{1noCO2}$ are identical to those described above in terms of solar constants and paleogeography but with a fixed atmospheric $CO_2$ concentration at pre-industrial levels (276.01 ppm). These simulations have been run for an additional 3000 years beyond those described in Valdes et al. (2021). To conduct this dust study, each of the 109 experiments is also run for an additional 30 years with the dust-relevant variables $(m, s, u, v)$ output at an hourly resolution.

The "0 Ma simulation" (the latest of the 109 simulations, corresponding to the pre-industrial configuration) is designed to be consistent with the rest of the paleo simulations in terms of most of the boundary conditions, for example, having homogeneous soil properties. As such, it is not the most accurate possible simulation of the pre-industrial climate. In order to tune the dust model with the pre-industrial scenario, we therefore also carry out another "standard pre-industrial simulation" with HadCM3L, which has more realistic pre-industrial boundary conditions. The main differences between them are – (1) the vegetation in the standard pre-industrial simulation is from observations, whereas the vegetation in the 0 Ma simulation is calculated by the coupled vegetation scheme; (2) the topography of the standard pre-industrial simulation is from observation, whereas the topography of the 0 Ma simulation is from the Scotese (2016) reconstructions; (3) the surface variables, primarily soil properties, are globally homogeneous in the 0 Ma simulation but are spatially varied based on observations in the standard pre-industrial simulation.

## 3 Dust model tuning

Given that there are uncertainties in the "correct" values of the thresholds $b_t$, $m_t$, $s_t$, $U_{t1}$, $U_{t2}$ and the coefficients $C_1$, $C_2$ etc., it is necessary to design a tuning process to identify the values that give the most realistic results for $D$. The assumption is that the tuning is carried out for the pre-industrial simulation, and the same tuned variables are assumed to be appropriate for all the paleo simulations.

### 3.1 Methods of tuning

In order to calibrate the dust model, a target needs to be selected as a benchmark. Theoretically, the target that the model is tuned to should be based on observations. In practice, the parameters commonly provided by modern dust observations are Dust Optical Depth or Aerosol Optical Depth (e.g. Gkikas et al., 2021), which are properties primarily influenced by the suspended dust particles in the atmosphere, whereas our dust model only predicts the surface emissions. Therefore, we use simulated modern dust emissions from other (more complex) coupled climate-dust models as the tuning target. Here we use the multi-model mean of 15 CMIP6 AMIP models (hereafter 15_MMM, Figure 1a) which is the dust emission over the period 2005-2014. In the absence of dust emissions from observations, this is justified partly by the complexity and resolution of CMIP6 models being higher than HadCM3L, which are therefore expected to predict more accurate dust emissions than the dust model in this study; and partly because the mean of several models in general has a lower bias than the output of a single model, and therefore represents a valid tuning target. The dust particle size range varies among the 15 models, with

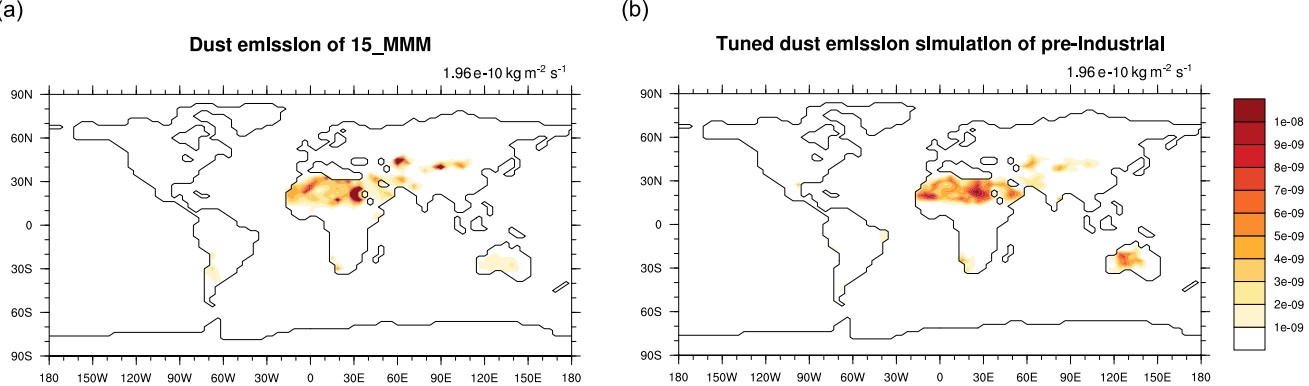

**Figure 1.** The pre-industrial dust emissions fields **(a)** averaged from the 15 CMIP6 AMIP models, which is used in this study as the tuning target, **(b)** simulated from the tuned DUSTY1.0 model (plot from model version one; see details about model versions in section 3.2 and Table 1). Both values at the top-right are the global mean dust emission rates.

the smallest size bin of 0.064-0.2 $\mu$ (HadGEM3-GC31-LL) and the largest with a diameter of 20-63 $\mu$ (HadGEM3-GC31-LL and UKESM1-0-LL) (Zhao et al., 2022). This also limits the range of particle sizes that DUSTY1.0 can effectively cover. The global average dust emission rate given by the 15_MMM is $1.96*10^{-10}\ kg\,m^{-2}s^{-1}$, which is used to calculate the $C_2$ in

205 equation (7) in each tuning experiment, so that the standard pre-industrial dust simulation has an identical global mean dust emission rate as the target. Every global mean value is calculated over all the grid boxes rather than over all the land grids to avoid confusion arising from variations in the total land area over time.

In addition to scaling the global average dust emission rate to the target, the main aim of the model tuning process is to find the optimum choices of the threshold values of the dust model that give the best fit to the spatial pattern of the target. When

210 evaluating the model, it is necessary to use a cost function to quantitatively measure the difference between the simulations and the target. The metric chosen here is the "Arcsin Mielke" score (AMS, Watterson et al. (2014)). For a modelled dust field D, and an observed dust field T, the AMS is defined as

$$AMS = \frac{2000}{\pi}arcsin[1 - mse/(V_D + V_T + (G_D - G_T)^2)] \tag{9}$$

where $mse$ is the mean-square error between $T$ and $D$, $V_D$ and $V_T$ are the spatial variances of $D$ and $T$ respectively, $G_D$

215 and $G_T$ are the spatial means of $D$ and $T$ respectively. The $AMS$ has a maximum possible value of 1,000, where the modelled result is identical to the target.

To perform the tuning, a Latin Hypercube Sampling (LHS, Mckay et al. (1979)) method is used. LHS is a stratified-random procedure which provides an efficient way of sampling variables. In this study, the number of samples, which is also the number of tested model versions, is 200. This decision is a trade-off between having as many samples as possible and the cost

220 of computing time. The five-dimensional parameter space where the 200 samples are taken from, is constrained by the range

of $b_t$, $m_t$, $s_t$, $U_{t1}$ and $U_{t2}$. We take a two-step approach here. Firstly, the initial range for each threshold is derived from the variable's distribution in the standard pre-industrial GCM simulation, or based on previous studies. Specifically, the bare soil values of the preindustrial simulations vary from 0 to 1, so the initial test range is set to be the [0.0-1.0]. The soil moisture at the southern edge of the Sahara area, where modern dust emissions are relatively low, is around 20.0 to 24.0 ($kg\ m^{-2}$), so the initial test range is set at a slightly wider range of 18 to 28 ($kg\ m^{-2}$). For snow cover, Lunt and Valdes (2002) apply a snow cover threshold in their dust model of 20.0 $kg\ m^{-2}$; as such the initial test range of 15.0 to 25.0 ($kg\ m^{-2}$) is chosen to encompass that value. The global minimum wind speed is around 0.8 $m\ s^{-1}$ and the maximum wind speed over land does not exceed 6.0 $m\ s^{-1}$, as such, a reasonable initial test range of $U_{t1}$ is set as 0.8 to 4.0 ($m\ s^{-1}$), and 4.0 to 6.0 ($m\ s^{-1}$) for $U_{t2}$. Secondly, starting from the initial test ranges described above, further adjusting (expanding, shrinking or shifting) to the test range has been carried out in an iterative process, with the aim of identifying appropriate ranges for the 200 sampled experiments. The final ranges applied are 0.2 to 0.34 for bare soil, 16.0 to 30.0 ($kg\ m^{-2}$) for soil moisture, 14.0 to 50.0 ($kg\ m^{-2}$) for snow cover, 0.2 to 2.0 ($m\ s^{-1}$) for $U_{t1}$ and 2.0 to 5.5 ($m\ s^{-1}$) for $U_{t2}$, as shown in Figure 2.

### 3.2 Tuning results

The results of the tuning exercise are shown in Figure 2. As a post hoc justification of the parameter ranges identified in Section 3.1, the results of the tuning show that the relatively high AMS values (i.e. the best fitting parameters) are not situated close to the edges of the ranges.

We select the dust emission model versions with the top five AMS results from the tuning exercise (as shown in black dots in Figure 2). The corresponding thresholds and coefficients are listed in Table 1. The tuned standard pre-industrial dust emission simulation (simulated from tuned parameter set version 1, all maps shown hereafter are results from model version 1 as a representative) is shown in Figure 1b. The tuned simulation generally captures the spatial pattern of global dust emission regions as defined by the 15_MMM, with northern Africa and central Australia predominating, followed by central Eurasia, and minor regions in South Africa. There are also many discrepancies exist between the tuned simulation and the target; for example, the high emissions hot spots in the Eastern Sahara and central Eurasia (Figure 1a) are not found in the stunned simulation (Figure 1b), while the tunned simulation gives higher emissions in the board Australia desert area. This discrepancy is largely due to the absence of detailed morphology and particle size representation in our models. These factors are crucial for capturing small-scale areas with frequent dust activation. Because we force the DUSTY1.0 model to match the global mean emission rate of the target, the average emission in each sub-area is enhanced in the absence of hot spots, which leads to overestimation in regions like Australia and western Sahara.

### 4 Phanerozoic dust emissions and controls

This section presents the paleo application of the dust model, including the simulated dust emissions over the Phanerozoic (Section 4.1), analysis of the contributions from the factors (Section 4.2), and the evolution of aridity in response to paleogeography (Section 4.3).

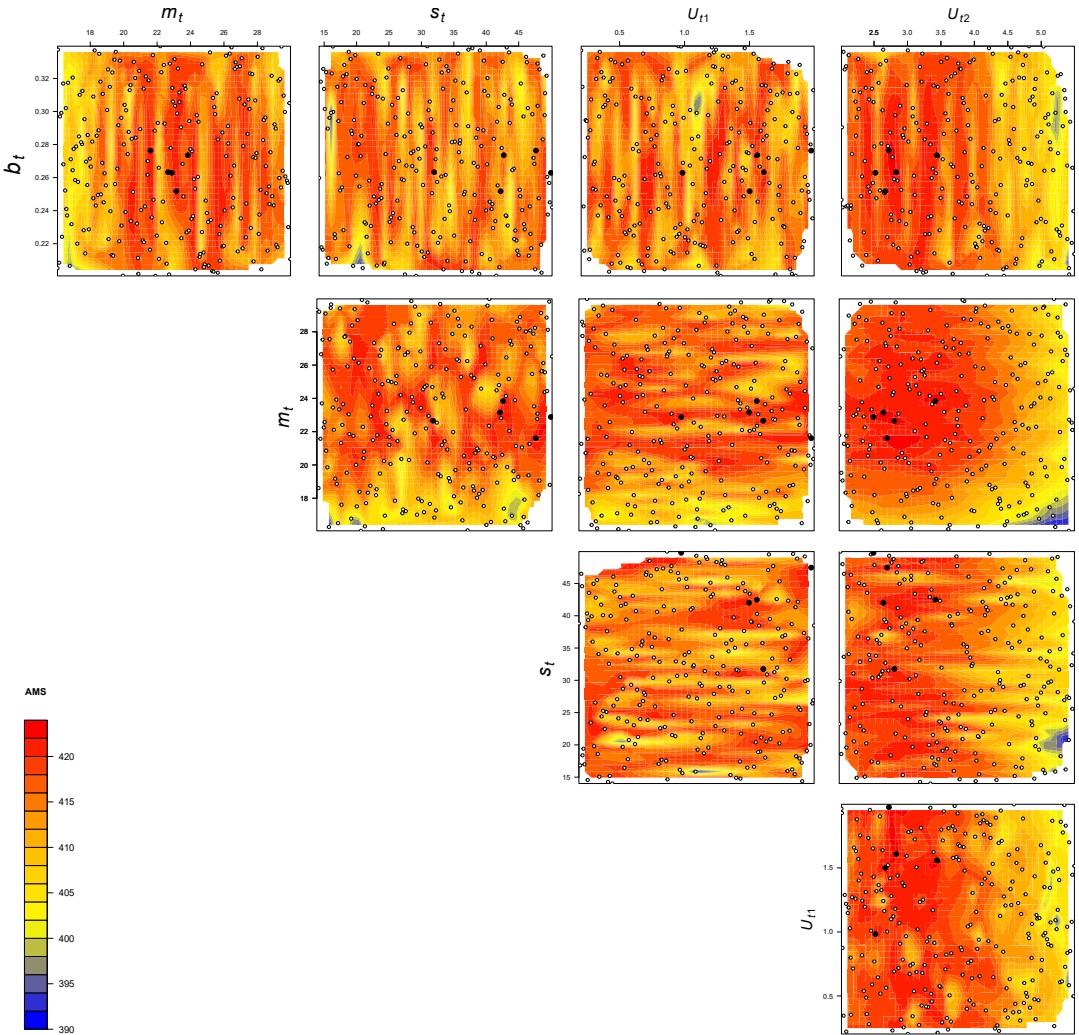

**Figure 2.** The performance of 200 dust emission model versions with different thresholds, measured by their AMS, compared to the tuning target. The ranges for each threshold shown in this figure are all final ranges, Each panel has 200 dots representing all the samples taken within the corresponding ranges, where the five solid dots are the top-ranked ones.

**Table 1.** Summary of thresholds valued applied in the top five DUSTY1.0 model versions

| Tuned model version number | $C_2$ | $b_t$ | $m_t$ | $s_t$ | $U_{t1}$ | $U_{t2}$ |
|---|---|---|---|---|---|---|
| 1 | 1.88e-08 | 0.26 | 22.66 | 31.75 | 1.61 | 2.83 |
| 2 | 2.11e-08 | 0.28 | 21.61 | 47.45 | 1.97 | 2.72 |
| 3 | 1.69e-08 | 0.27 | 23.84 | 42.49 | 1.56 | 3.44 |
| 4 | 1.78e-08 | 0.25 | 23.16 | 42.00 | 1.50 | 2.67 |
| 5 | 1.83e-08 | 0.26 | 22.88 | 49.76 | 0.98 | 2.52 |

## 4.1 Dust emissions over the Phanerozoic

The five tuned versions of the dust emission model in section 3.2 are applied to all 109 simulations from series $S_2$ (see details in section 2.3 through the Phanerozoic (since 540 Ma). Here we show the simulated global averaged dust emission rate time series in Figure 3, and the simulated dust emission fields for five example time slices (0 Ma, 196Ma, 252Ma, 370Ma, and 530 Ma) in Figure 4 (the dust emission field for all 109 experiments can be found in Appendix in Figure A2-A5). Most notably, the dust emissions are significantly high during the period over the late Permian to the early Jurassic (263 Ma to 191 Ma), where there are two peaks, one covering the Permian-Triassic transition (252 Ma) at the emission rate of $9.1 * 10^{-10} \, kg \, m^{-2} s^{-1}$ and the other spanning the middle Triassic to the early Jurassic (196 Ma) at the emission rate of $8.5 * 10^{-10} \, kg \, m^{-2} s^{-1}$. The emission rates for both peaks are more than four times the pre-industrial level ($1.96 * 10^{-10} \, kg \, m^{-2} s^{-1}$). The dust provenance areas for both peaks are concentrated in the central supercontinent, covering the mid-latitudes, subtropics, and the tropics, as illustrated in Figure 4b and c. The global dust emission rate reaches its lowest in the Devonian-Carboniferous transition (366 Ma) at the level of $1.9 * 10^{-11} \, kg \, m^{-2} s^{-1}$, which is almost ten times less than the pre-industrial level. Dust provenance areas are scattered throughout the mid-latitudes in both hemispheres (Figure 4d) at that time. In the early Cambrian (541 Ma to 535 Ma), the oldest simulation we carry out, dust emission rates ($1.6 * 10^{-10} \, kg \, m^{-2} s^{-1}$) are close to the pre-industrial level, and the dust source regions are centred in the mid-latitudes of both hemispheres (Figure 4e). Evaluations of the simulated dust emission in comparison to geological records will be discussed in section 5.1. Similar to the discrepancy mentioned in section 3.2, our paleo simulations may not be able to represent high-emission hot spots and, alternatively, may overestimate emission rates in broadly-distributed but low-emission-rate desert areas. Further uncertainties arising from the GCM simulating vegetation in the absence of land plants during the early Paleozoic will be discussed in section 5.2.

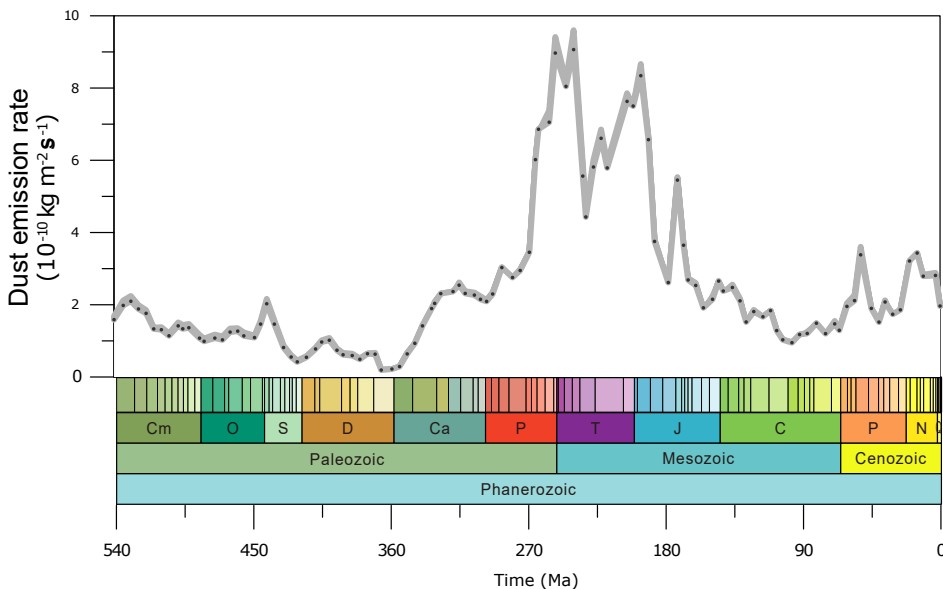

**Figure 3.** Time series of the simulated global average dust emission rate over the Phanerozoic. The grey shading (not a curve) in the figure is a stack of the results from those five top-ranked model versions, and the dots on it represent the results of model version one to hint at the time intervals between simulations of different time slices.

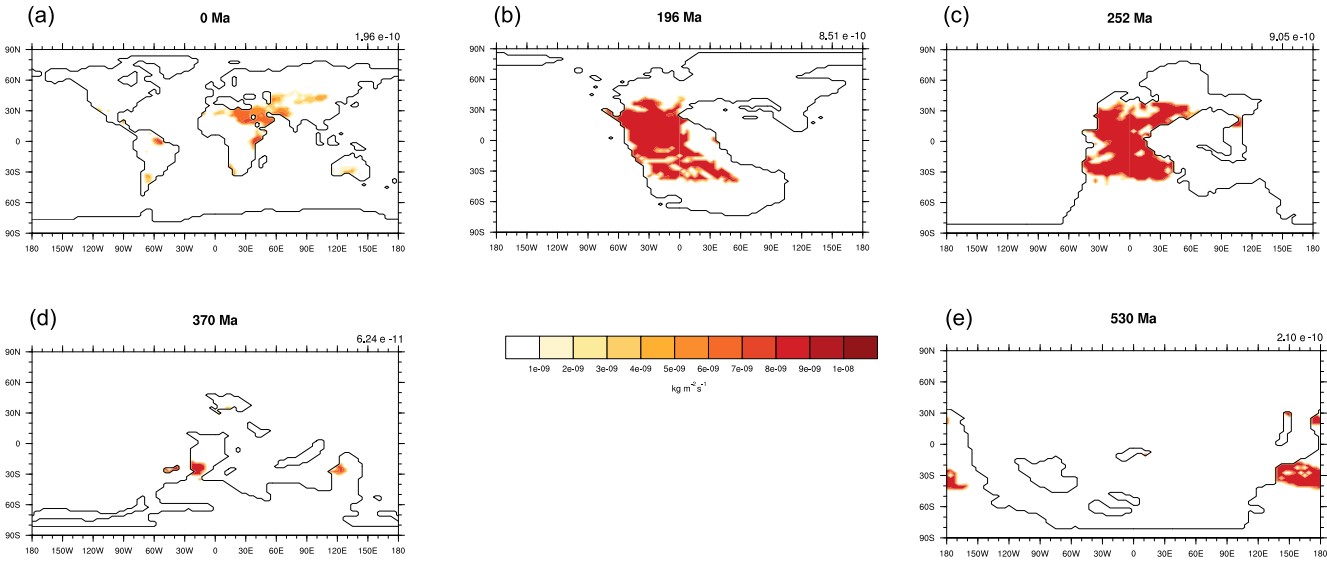

**Figure 4.** A few examples of the simulated dust emissions. The examples are for **(a)** present day (0 Ma), **(b)** the early Jurassic (196 Ma), **(c)** the early Triassic (252 Ma), **(d)** the late Devonian (370 Ma), and **(e)** the early Cambrian (530 Ma). Values at the top-right of each panel are the corresponding global mean dust emission rate. All unites are $kg\ m^{-2}s^{-1}$.

## 4.2 Contributing factors to the dust emissions

The simulated dust emissions described in section 4.1 show substantial fluctuations over the Phanerozoic. In order to understand the causes of the fluctuations, we here use a factorisation approach to analyze and quantify the relative importance of each variable (the land-sea distribution $l$, the non-vegetated area $b$, the soil moisture $m$, the snow cover $s$ and the near-surface wind $U$, all introduced in section 2.1) that is used to drive the dust emission model.

The factorisation is carried out in a two-step approach, where the analysis of $l$ and $b$ is performed as a linear factorisation, whereas the analysis of $m$, $s$ and $U$ is performed based on the 'linear sum' factorisation method according to (Lunt et al., 2021), which is suitable for diagnosing a multi-variate system. We understand the whole system from a starting point in which there are no dust emissions anywhere on the globe, $D^0$. We then add the factors $l$ and $b$ sequentially, to produce $D^l$ and $D^{lb}$ respectively (the linear factorisation). We then carry out an unordered addition of the three remaining factors $m$, $s$, and $U$, producing $D^{lbm}$, $D^{lbs}$, $D^{lbU}$, $D^{lbms}$, $D^{lbmU}$, $D^{lbsU}$ and $D^{lbmsU}$ (see Section 2.1 for the naming convention). Details of the factorisation are described in the appendix with the factorisation concept illustrated in Figure A1. The contributions of each factor are quantified for the land-sea distribution, vegetation, soil moisture, snow cover, and wind, as $\Delta D^l$ for the land sea distribution, $\Delta D^b$,$\Delta D^m$,$\Delta D^s$, and $\Delta D^U$, respectively.

The factorisation is applied to all the time slices to quantify the contribution of each factor and how this varies over time, as illustrated in Figure 5. The contribution from the land-sea distribution factor ($\Delta D^l = D^l$, brown line in Figure 5a) results in dust emissions of about 3.5 $kg/m^2$ over the early Phanerozoic, increasing to a maximum around 250 Ma, and then decreasing to about 90 Ma before increasing slightly to the modern. Following is the contribution from the non-vegetated area factor ($\Delta D^b$), which reduces the dust emissions remarkably over time. As expected, the non-vegetated area and land-sea distributions are very important, and the joint contribution from both $l$ and $b$ ($\Delta D^{lb}$) gives a more concrete reflection of this. It shows that they are three times the all-forcing dust emissions ($D^{lbmsU}$). The addition of $m$ and $s$ both reduce the dust emission. The contribution from $m$ ($\Delta D^m$) is approximately double the all-forcing dust emissions, whereas the contribution from $s$ ($\Delta D^s$) is only 1% of the all-forcing dust emissions. The addition of $U$ alters dust emissions most positively over time but also has adverse effects for a few time periods in the mid-Permian, early-Cretaceous and late-Paleogene. It contributes an average of 44% of the all-forcing dust emissions before the Carboniferous and 27% of the all-forcing dust emissions during the late Permian to the early Jurassic.

The analysis above quantifies the magnitude of each factor to the total global dust emissions for each time slice individually. A complementary approach is to quantify the impact of each factor on the temporal variability of emissions throughout time, i.e., the extent to which the shape of the emission timeseries improves towards the all-forcing emission timeseries as the result of the inclusion of a specific variable. In order to evaluate this, for each combination of factors, we calculate the correlation coefficient of the resulting emissions timeseries with the all-forcing emissions timeseries. We then drive the contribution from the difference in these correlation coefficients, The greatest contribution to the temporal variability in emissions is made by $l$ (66.53%), followed by $b$ (31.86%), $U$ (1.22%), $m$ (0.34%), and finally $s$ (0.05%).

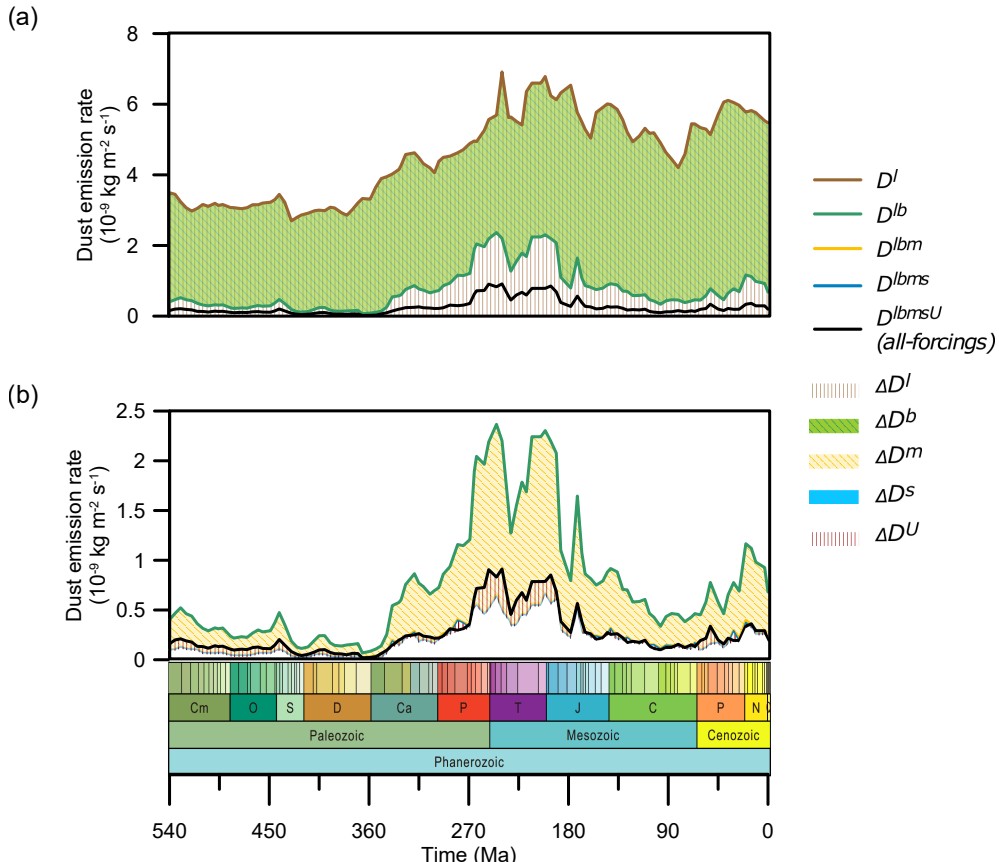

**Figure 5.** The decomposed contribution of individual factors to the total dust emissions over the Phanerozoic. **(a)** Results for $l$ and $b$,**(b)** Results for $m$, $s$ and $U$. Note that the ranges of y-axes differ between panel **(a)** and **(b)**. All curves refer to the joint dust emissions derived from the corresponding superscript variables, and the shadings refer to the dust emissions contributed from each specific superscript variable. (e.g. $D^{lb}$ refers to the simulated dust emissions when only variables $l$ and $b$ are included in the DUSTY1.0 model, and $\Delta D^m$ refers to the quantified dust emissions contribution of the sole variable $m$)

## 4.3 The paleogeographical control on the desert area

305 As shown in Section 4.2, the combined land and non-vegetated area factors ($D^{lb}$) dominate the dust emission variations on geological timescales. Given that the non-vegetated area changes implicitly also include the land area changes (i.e. $D^b=D^{lb}$), here we focus on the mechanisms for the changes in non-vegetated areas.

We here measure the extent of aridity with an aridity index ($AI$) that is threshold-based and inversely proportional to the amount of precipitation:

310 $$AI = (t_{arid} - precipitation)/t_{arid}, \tag{10}$$

where $t_{arid}$ is set as 0.8 mm/day. This threshold is selected because sediments such as evaporite and weathered sandstone, which are indicators of arid environments, are formed under a limit of approximately 0.8 mm/day (Cecil, 2003; Warren, 2010; Price et al., 1995).

The non-vegetated fraction in the simulations are very closely controlled by the precipitation rate. In particular, the area of precipitation that is less than 0.8 mm/day (represented by the Aridity Index) shows a very strong positive linear relationship with the non-vegetated area (Figure 6ab, correlation coefficient = 0.98). And as we have shown above, the non-vegetated fraction controls the total dust emissions. In addition to the global means, the zonal mean distribution of continental precipitation varies over time (Figure 7c), as does the zonal mean Aridity Index (Figure 7d. The aridity index shows a very similar pattern to the dust emissions (Figure 7e). As such, the continental distribution, combined with the zonal mean precipitation (Figure 7ab) are sufficient to closely approximate the dust emissions (Figure 7e).

The challenge then becomes to understand the controls on zonal mean precipitation (Figure 7a). There are three boundary conditions that are changing over time which must ultimately control the precipitation changes; namely, $CO_2$, solar constant, and paleogeography. Firstly, in order to explore the impact of $CO_2$ change, here we take the results from an additional series of GCM simulations, as a sensitivity test. Comparison of results from the $S_1$ and $S_{1noCO2}$ simulations (described in section 2.3) reveal the negligible effect of $CO_2$ concentration on the low precipitation area variation over time (Figure 8b), and therefore aridity (Figure 8a). Hence, we conclude that on these timescales, $CO_2$ is not a strong control of the aridity. Furthermore, the change in solar constant through the Phanerozoic is linear, whereas the dust emissions change is highly non-linear. As such, we can attribute the mechanism of changes in precipitation, aridity, and ultimately dust emissions, to the only remaining boundary condition change, namely paleogeography.

In general, it may be expected that regions in the continental interiors are more arid due to being remote to oceanic moisture sources. Here we develop indices measuring the distance of the shortest pathway from land to ocean. The distance indices are calculated in the tropical zone (23.5°N to 23.5°S, $TD$), the subtropical zones in the Northern Hemisphere (35°N to 23.5°N, $NSTD$) and the subtropical zones in the Southern Hemisphere (23.5 °S to 35°S, $SSTD$) respectively, as shown in Figure 6c. During the period prior to the Carboniferous, the global aridity (Figure 6b) shows a remarkably similar temporal evolution to $TD$ and $SSTD$, which drop remarkably in the Silurian and Devonian. Since the Carboniferous, more continents shift to the tropics and the Northern Hemisphere, and the overall growth of global aridity is therefore driven by the combined effect of $TD$ and $NSTD$ until the late Permian (excepting a short drop-down at the late Carboniferous when the $SSTD$ exerts a control). The extreme aridity from the late Permian to the late Triassic is controlled by all three land-sea distance indices, which is due to the fact that the supercontinent covers almost all latitudes during the corresponding times. The extreme aridity can be divided into two peaks during this period. The first aridity extreme at the Permian-Triassic boundary is mainly driven by the $SSTD$, the second aridity extreme at the late Triassic is mainly driven by the $TD$, while the $NSTD$ contributes equally to both. The land-sea distance indices have decreased in all three latitude zones since the early Jurassic due to continuous continent convergence, resulting in overall humidification until the Cretaceous-Paleocene boundary, where the decrease of $SSTD$ plays a relatively significant role. It is important to note that there are other factors contributing to this widespread aridity, which corresponds to the collapse of the tropical land humid zones as shown in Figure 7c. With the latest improvement

in vegetation albedo parameterization in the GCM (see description in section 2.2), the low vegetation coverage will trigger a positive feedback loop where higher surface albedo leads to lower net solar radiation and an increase in radiative cooling. This then suppresses convection and leads to amplification and potential expansion of the dry conditions (Charney et al., 1977). From the Cenozoic onward, the continents have shifted further northward, resulting in the $NSTD$ being the dominant control of the expanding aridity.

Overall, as the fundamental driving factor, paleogeography controls the geological timescale aridity and, thus, desert area variations over the Phanerozoic.The desert dust is consistently found in subtropical land areas of either hemisphere over time and also in the tropical during supercontinent times. These can be mainly explained by variations in the land-to-sea distance. To some extent, the higher albedo over sparesely vegetated areas further amplifies the spreading of aridity into the tropics during supercontinent times.

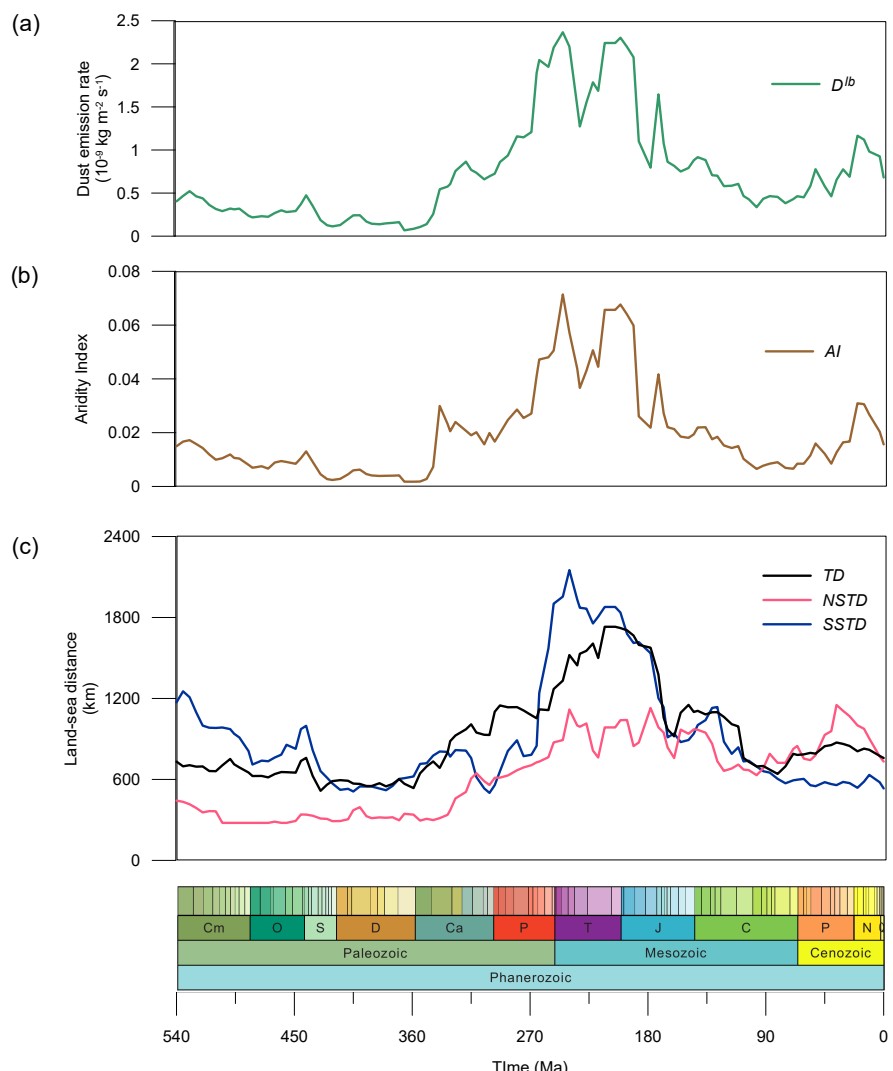

**Figure 6.** Time series of **(a)** $D^{lb}$, which is the simulated global average dust emissions jointly contributed from $l$ and $b$, **(b)** the AI (aridity index), which reflects the lowness of precipitation, and **(c)** the land-sea distance indices, in which $TD$ refers to the tropics, $NSTD$ and $SSTD$ corresponds to the subtropics of the Northern and Southern Hemispheres, respectively.

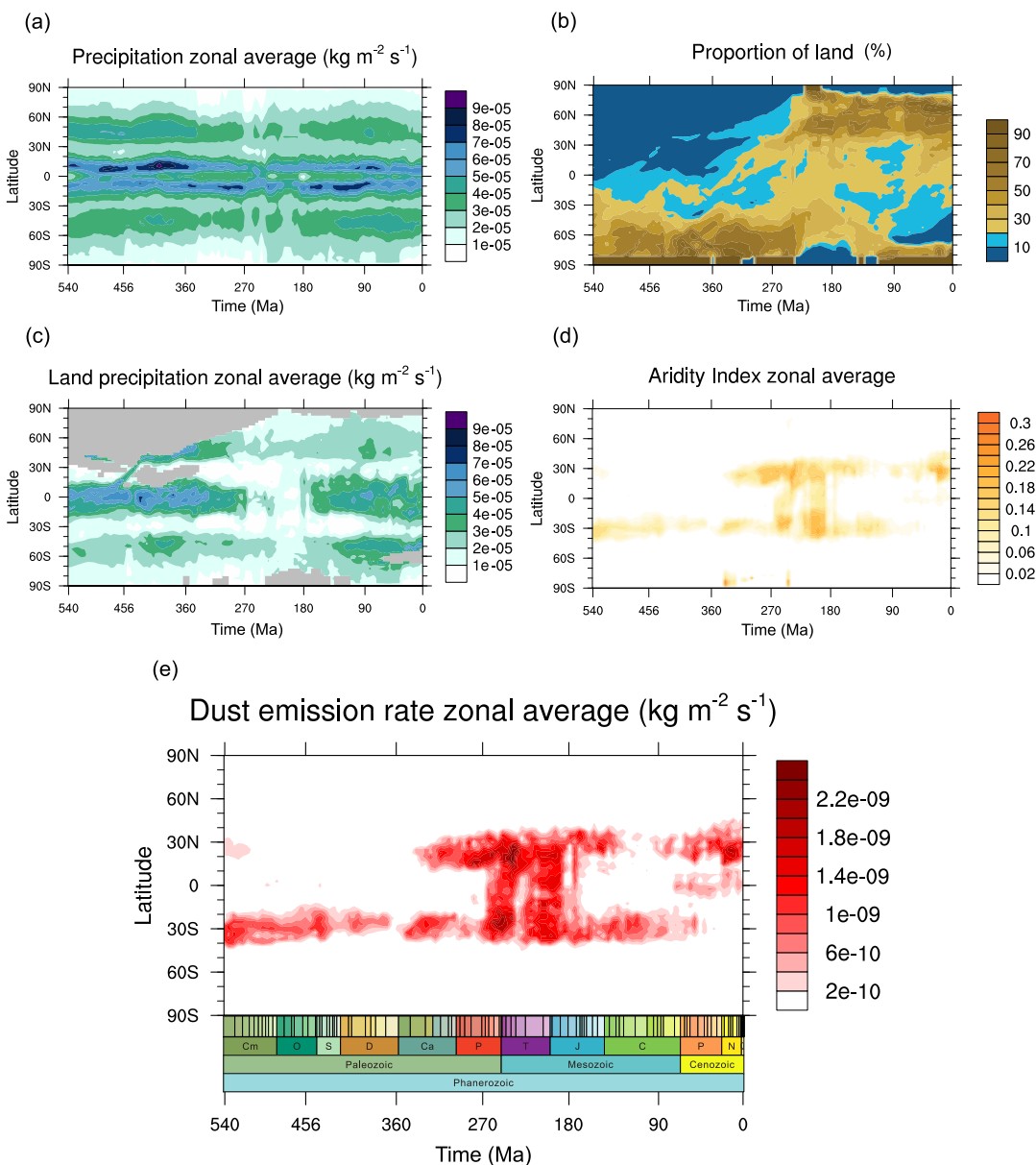

**Figure 7.** The zonal means of **(a)** the simulated precipitation, **(b)** the proportion of land , **(c)** the simulated precipitation over land, **(d)** the aridity index, **(e)** the simulated dust emissions over the Phanerozoic.

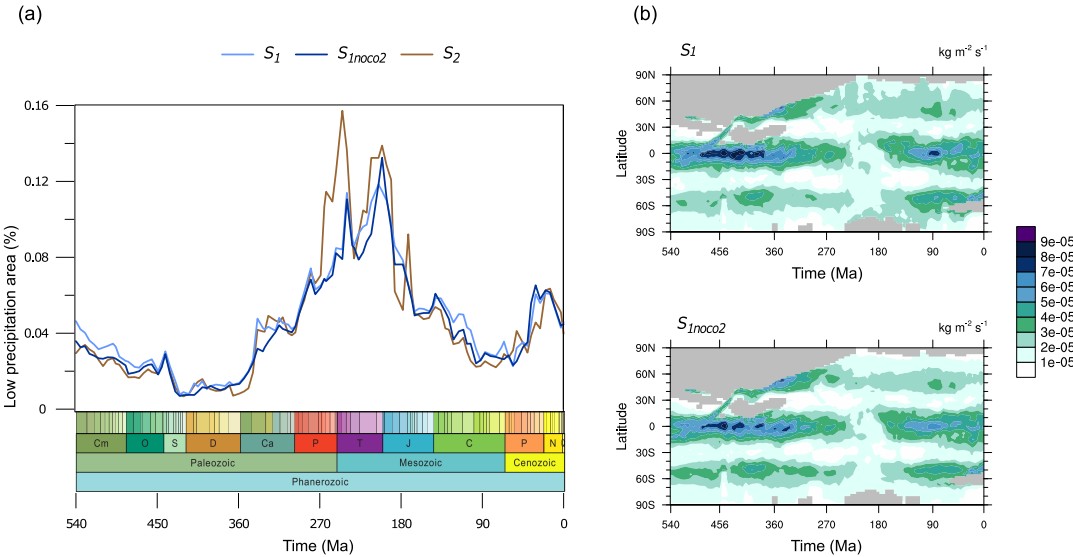

**Figure 8.** **(a)** Comparison of low precipitation area fraction time series from three different experimental series: $S_1$, $S_1noco2$ and $S_2$. A daily mean precipitation of less than 0.8 mm is considered low precipitation. **(b)** Comparison of zonal mean land precipitation over the Phanerozoic between experimental series $S_1$ and $S_1noco2$.

## 5 Discussion

### 5.1 Credibility of the simulated Phanerozoic dust

We herein compare evidence from both sediments and modelling to evaluate the simulated Phanerozoic dust emissions.

The evaporite sediment indicators compilation from Boucot et al. (2013) is used here to assess our simulation. This model-data comparison approach is indirect because evaporite is not a proxy for dust emission; however, it is chosen due to the absence of an appropriate proxy for dust emission, especially in deep-time paleo contexts. As a compromise, we compare the evaporite sediment records to the HadCM3L simulated evaporite, effectively evaluating the performance of the GCM in simulating the paleo hydrological cycles. The GCM predicts evaporation using the following criteria: mean annual precipitation less than 1400 mm, no more than 3 months with precipitation greater than 40 mm, and 6 months or more with mean temperature exceeding 20 °C (Craggs et al., 2012, scheme A). Similar evaporite prediction schemes are also used in Price et al. (1995); Bao et al. (2023) although they uses different criteria. This scheme roughly represents a rather arid environment in terms of its constraints on precipitation. Although the scheme is also temperature dependent, as has been argued in section 4.3 that atmospheric $CO_2$ induced hydroclimate change is neglectable, also considering that $CO_2$ is considered to be primarily driving the temperature change over the Phanerozoic (e.g. Mills et al., 2019; Royer et al., 2004), we ignore the effect from the temperature on evaporites predictions in the context of our focus on the hydrological cycles over the Phanerozoic timescale. This model-data comparison also serves as a side evaluation of the DUSTY1.0 model because the results from this study indicate that the dust emission is primarily dominated by the aridity (i.e. hydrological cycles) over the Phanerozoic (section 4.3).

How well the model matches with data is evaluated by how well the evaporite data in the compilation agree with estimates of potential evaporite distribution derived from the HadCM3L simulations. Evaporite data points from Boucot et al. (2013) are rotated back to their paleo coordinates. We do not include data points that are outside the land mask in our paleogeograohy configuration, because the formation mechanism of those evaporites found on the coastline and continental shelf are mainly related to the marine incursion and thus do not represent arid paleoenvironment.

The performance of the model, indicated by the overlay ratio of evaporite data points on the model-predicted evaporite areas, varies over time. On average, 35.3% of the evaporite data points are correctly predicted by the HadCM3L model over the Phanerozoic. To put this agreement in content, we also derive a "zero skill model", in which the fixed evaporite simulations for 0 Ma are applied for all paleo timeslices, constraining the lowest performance limit from a random model prediction. The zero-skill model achieves an average agreement of 4.6%, significantly lower than the HadCM3L model's performance, indicating the adequate accuracy of the model used in this study (Figure 9e). Notably, the simulated widespread tropical drought during the late Permian to early Jurassic, which leads to widely distributed potential evaporites, corresponds fairly well with the geological sedimentary evidence (Figure 9ab), with an averaged overlay ratio of 67.7%.

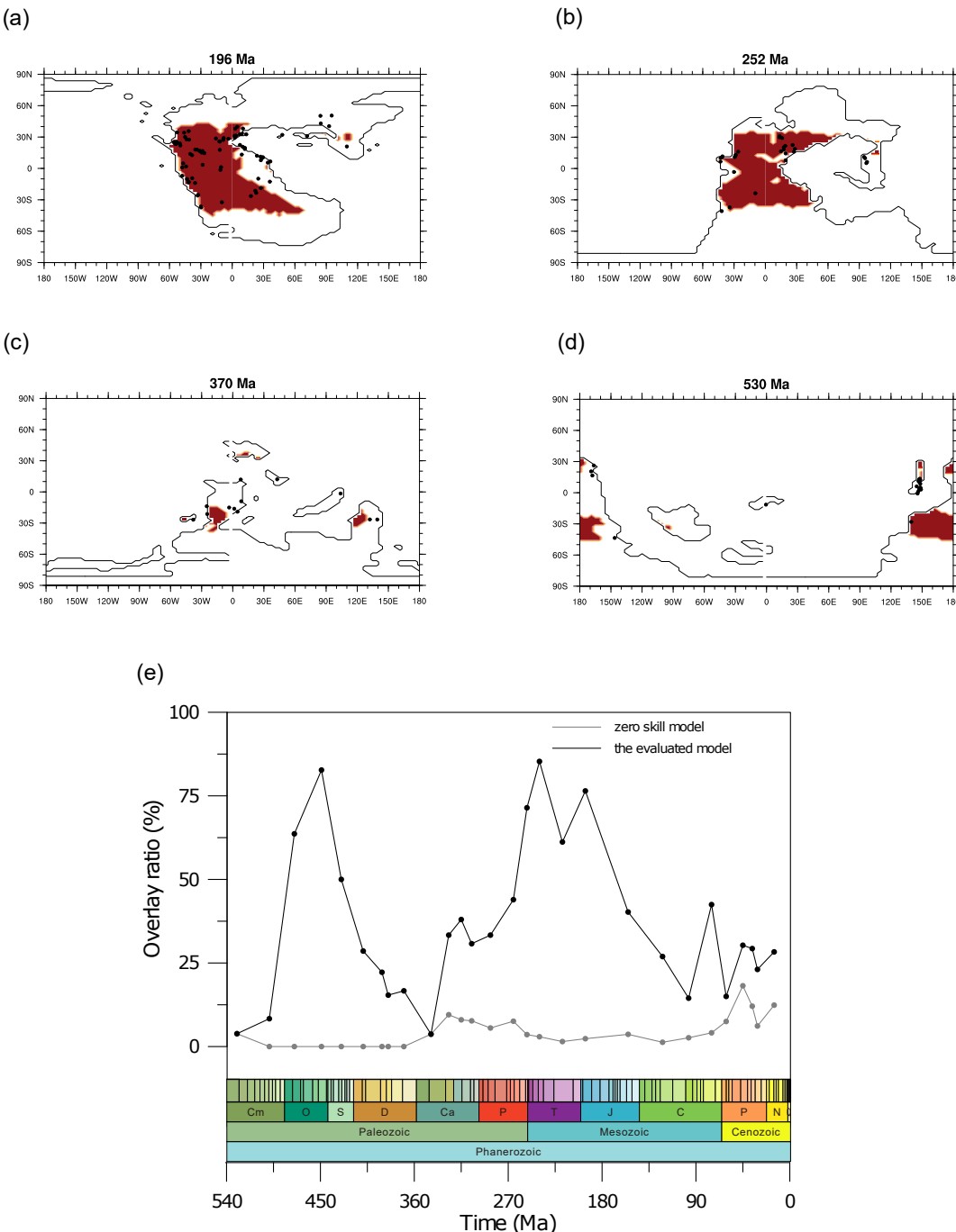

**Figure 9.** Comparisons of evaporite sedimentary records (dark dots, data from (Boucot et al., 2013)) to the simulated evaporite (area in red) of a few example time slices: **(a)** the early Jurassic (196 Ma), **(b)** the early Triassic (252 Ma), **(c)** the late Devonian (370 Ma), **(d)** the early Cambrian (530 Ma), and **(e)** a time series representing the performance of the model derived by assessing the overlap between model and data over time.

Here, we further focus on the Mesozoic to Cenozoic Asian desert dust and the Mesozoic Pangean desert dust to evaluate the accuracy of our simulations.

Our results show that the Asian desert dust generated since the middle Jurassic (168 Ma), reached a maximum at the Late Jurassic–Early Cretaceous (145 Ma). During the early Cretaceous (131 - 97 Ma), the Asian continent passed through a period without obvious desert dust. The dust provenance then became very broad in subtropical Asia from the late Eocene (36 Ma) to the present. As documented in previous studies, the extreme Asian aridity evidenced from formations in the Junggar Basin of the Late Jurassic–Early Cretaceous (Jolivet et al., 2017) is well simulated from our results. Results from previous modelling studies show that central and western Asia was generally dry at the end of the Late Cretaceous, but was not identified as unvegetated dust provenance (Zhang et al., 2019). our results constrain the arid dust emission region to the central and west subtropical belt, which is also accordant to the proxy evidence of eolian dunes recorded in the earlier late Cretaceous (Farnsworth et al., 2019; Hasegawa et al., 2012). The permanent aridification recorded in the present-day Xining Basin originated from the late Eocene to Oligocene (40 Ma) (Licht et al., 2016; Bosboom et al., 2014), our simulations show the dust provenance appears since 36 Ma, which is in good agreement.

The dust emissions during the late Permian to the early Jurassic, as identified in this study, are remarkably high over the whole Phanerozoic. Our results suggest that the extensive dust emissions across the tropical and subtropical supercontinent specifically occurred from 265 Ma to 190 Ma. Flora biome evidence and a general lack of plant fossil record (Nowak et al., 2020) supports the massive desert environment which was distributed almost identically to our simulated dust provenance during the late Permian (256 Ma). Sediment records (Boucot et al., 2013) indicate similar massive aridity patterns as our simulations – the sediments reflecting wet and semi-arid environments in the central tropical land were not documented while the sediments reflecting arid environments were abundantly found along the edges of tropical and subtropical land since the Middle-Late Permian (265 Ma) to the middle Triassic (240 Ma) (Figure 9), suggesting a massive extreme arid extended from subtropic to tropic during the corresponding times.

## 5.2 Implication and Limitation

Our study for the first time presents the dust emission variations over the past 540 Million years with the dust emission fields for each of the 109 corresponding time slices, which can serve as an archive both in the field of dust simulations and paleoclimate. Similar simulations focusing on deep-time dust variations have been performed by Lin et al. (2024) with a different GCM and vegetation scheme. Our results in general support their conclusion that subtropical land is the major controlling factor in the dust variations since the late Permian, and extend this mechanism to the whole Phanerozoic with higher temporal resolution experiments. Further model-data comparison and model-model comparison can be carried out on this basis in the future to improve the understanding of deep-time dust variations and their mechanisms.

Some of the bias from our results may arise from details about the dust emission process that have to be omitted in this study. As mentioned in section 2.1, the particle size and topography parameters are significant to the physical dust emission process but are not included here due to a lack of information.

Uncertainties in our simulated results could have been caused by the vegetation simulations in different ways. We here in this study assumed plant functional types are consistent and the same as modern over the whole Phanerozoic with applying the TRIFFID interactive vegetation model to all paleo time slices. This will theoretically result in inappropriate vegetation outputs, especially in the older time results. To constrain the uncertainty during the time before land plant colonization, we carry out sensitive tests by running the DUSTY1.0 model with the assumption that all land areas are non-vegetated areas, except for those prescribed as ice sheets, prior to the end of Devonian (360 Ma) (Figure 10). Sensitivity tests reveal that results shown in section 4.1 underestimate the dust emissions over the early-Paleozoic times, with a factor of doubling underestimate at the Ordovician-Silurian transition emission peak and about two times underestimate at the early Devonian emission peak, whereas for most of the other early Paleozoic times, the level of underestimation is mostly within the range of 5% to 30%. The degree of underestimation is not as significant as the decomposed contributions from the non-vegetated area $b$ derived in section 4.2. This is because of the coupled variations between vegetation and soil moisture. Even with the assumption of no land vegetation, the simulated areas of desert dust are similarly constrained by the soil moisture. However, there is a bias in these estimations of uncertainty because the dynamic interactions between vegetation and climate are neglected, which possibly explains the discrepancy compared to the up to fourfold underestimation in the Precambrian (Liu et al., 2020). A more accurate evaluation of underestimation could be done by running the GCM simulations without the TRIFFID scheme. Due to the matter of computational cost, this is not performed within this study. Moreover, the lack of vegetation would likely also result in a very different land surface during these times, which may also have a big impact on emission properties. Our simulations after the Devonian ( 360 Ma) are more robust than those prior to that when terrestrial flora is more similar to today. While this constant vegetation scheme is the common approach of most current paleoclimate modelling studies, the new trait-based whole-plant functional-strategy approach (Matthaeus et al., 2023) shows potential to be an alternative solution. The scope of this study has focused on only looking at the non-vegetated area rather than a specific vegetation type, despite the unavoidable uncertainties coming with the vegetation scheme, We argue that the modelled dust and aridity are relatively robust, as any vegetation may have flourished where there was precipitation.

Another aspect of uncertainties caused by the vegetation scheme is related to the atmospheric $CO_2$. The vegetation model TRIFFID predicts plant functional types by taking atmospheric $CO_2$ in addition to temperature and precipitation from the GCM, and a higher $CO_2$ will fertilize the vegetation. As such, we emphasise the uncertainties of our simulated dust emissions, presumably underestimations, for the periods before 420 Ma, when the CO2 forcings are derived from an "extension" of the Foster et al. (2017) data and appear very high (Valdes et al., 2020). The explicit effects of $CO_2$ on vegetation could be explored by running the vegetation model offline with consistent temperature and precipitation but different $CO_2$, which could be tested in future research.

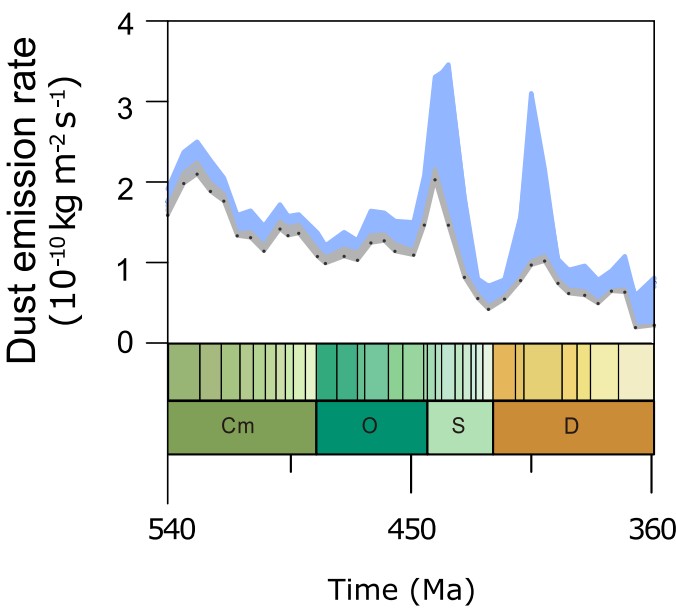

**Figure 10.** The uncertainty of dust emission rates tested by assuming there is no land plant prior to the end of Devonian, as shown in blue shading. The grey shading is the simulated dust emissions identical to those illustrated in Figure 3.

In this study, the dust emission model does not give a complete dust cycle, nor is it fully coupled to the GCM, so the dust feedback is not represented in the simulations. The effect of dust on the corresponding paleo climate could have been significant, especially for periods with enormous emissions. For the time periods where our results show high dust emissions, the radiation effect of dust may be very different from the pre-industrial time or the Last Glacial Maximum. The latter two are relatively well constrained, whereas the effects in high-emission deeper times require further diagnosis. There are many models which include a dust scheme (e.g. the CMIP6 models evaluated in Zhao et al. (2022), but would take huge computational resources to run simulations of similar temporal resolution due to their complexity, while offline dust models are more suitable for carrying out multi-scale simulations with the defect of losing dynamic feedback between the dust cycle and the climate system. In addition, these models often require much greater detail of dust input parameters that is possible for the deep past.

Another aspect of limitation is the model designed for this study is not supposed to test the hypothesis related to the sediment supply process, e.g. the hydrological basins and glaciogenic dust source, as described in section 2.1. These explorations could be pursued in future research with higher computational resources, using more complex models including those parameterizations.

## 6   Conclusions

In this study, we simulate dust emissions over the Phanerozoic with an offline dust emission model DUSTY1.0 and the General Circulation Model HadCM3L. The dust emission model is first tuned to the modern and then applied to the paleo configurations.

For the first time, our study yields a timeseries of global dust emission rates of the whole Phanerozoic, along with corresponding dust emission fields at the stage level resolution. The early Paleozoic may have underestimated simulated dust emissions before land plant colonization. Model-data comparisons reveal that our simulations are reasonable in accordance with geological records.

The driving forces of dust emissions over time are analysed at different levels of mechanisms. From the perspective of dust emission-related surface processes, our results suggest the non-vegetated area is the predominant contributing factor to the multi-million-year-scale dust emissions variations, both in magnitude and in patterns of changes versus time. From the perspective of the geological timescale, our results suggest that paleogeography dominates the variations in continent aridity and, therefore, dust emissions, while the effect from $CO_2$ is negligible.

*Code and data availability.* The DUSTY1.0 model code is available and the simulated dust emissions fields are available from https://github. com/yixuan-coding/DUSTY1.0

## Appendix A: Appendix

Here we provide the detailed methods of factorisation used in section 4.2.

For the linear factorisation on $l$ and $b$. The dust emissions contributed by $l$ ($\Delta D^l$) is calculated by:

$$\Delta D^l = D^l \qquad \text{(A1)}$$

representing the dust emissions difference between the configuration in which there is no dust emission over the whole globe, and the configuration in which all the land area produces dust. The corresponding states of other variables in the configuration in which all land is vegetation-free are: $b = 1$, which means all the land surface is covered by 100% bare soil; $m = 0$, which means the surface soil is at its driest over all land; $s = 0$, which means there is no snow cover anywhere; $U = 1.91$ (in the choice of tuned parameter set version 1), indicating the surface wind speed that blows up the dust. Similarly, the dust emissions contributed by $b$ ($\Delta D^b$) is calculated by:

$$\Delta D^b = D^{lb} - D^l, \qquad \text{(A2)}$$

which represents the difference in dust emission rate between the configuration in which all land is vegetation-free and the configuration in which dust sources are restricted to non-vegetated land areas. The corresponding states of other variables in this configuration are identical to those in the all land is vegetation-free configuration except that $b$ is assigned specific values.

For the non-linear factorisation on $d_m, d_s$ and $d_U$, the algorithm is as follows according to Lunt et al. (2021) to calculate the difference between $D$ and $\Delta D^{lb}$:

The resulting equations for the factorisations are :

$$\Delta D^m = \frac{1}{6}\left[2(D^{lbm} - D^{lb}) + (D^{lbmU} - D^{lbU}) + (D^{lbms} - D^{lbs}) + 2(D^{lbmsU} - D^{lbsU})\right]$$

$$\Delta D^s = \frac{1}{6}\left[2(D^{lbs} - D^{lb}) + (D^{lbms} - D^{lbm}) + (D^{lbsU} - D^{lbU}) + 2(D^{lbmsU} - D^{lbU})\right] \tag{A3}$$

$$\Delta D^U = \frac{1}{6}\left[2(D^{lbU} - D^{lb}) + (D^{lbmU} - D^{lbm}) + (D^{lbsU} - D^{lbs}) + 2(D^{lbmsU} - D^{lbms})\right]$$

where the $\Delta D^m$, $\Delta D^s$ and $\Delta D^U$ represent the difference in dust emission caused by the $m, s$ and $U$ from the "all-bare-
land-desert" configuration respectively.

The total dust emission rate is hence isolated as:

$$D = \Delta D^l + \Delta D^b + \Delta D^m + \Delta D^s + \Delta D^U \tag{A4}$$

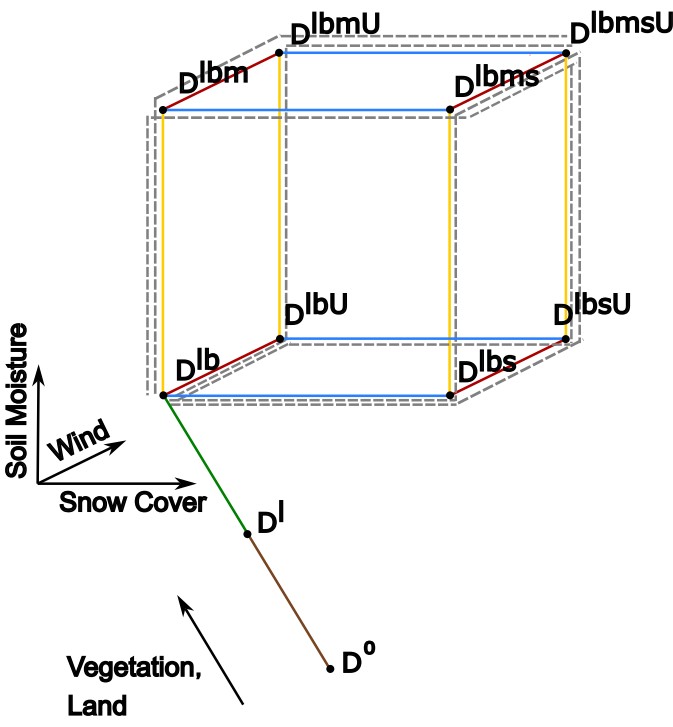

**Figure A1.** The factorisation method concept

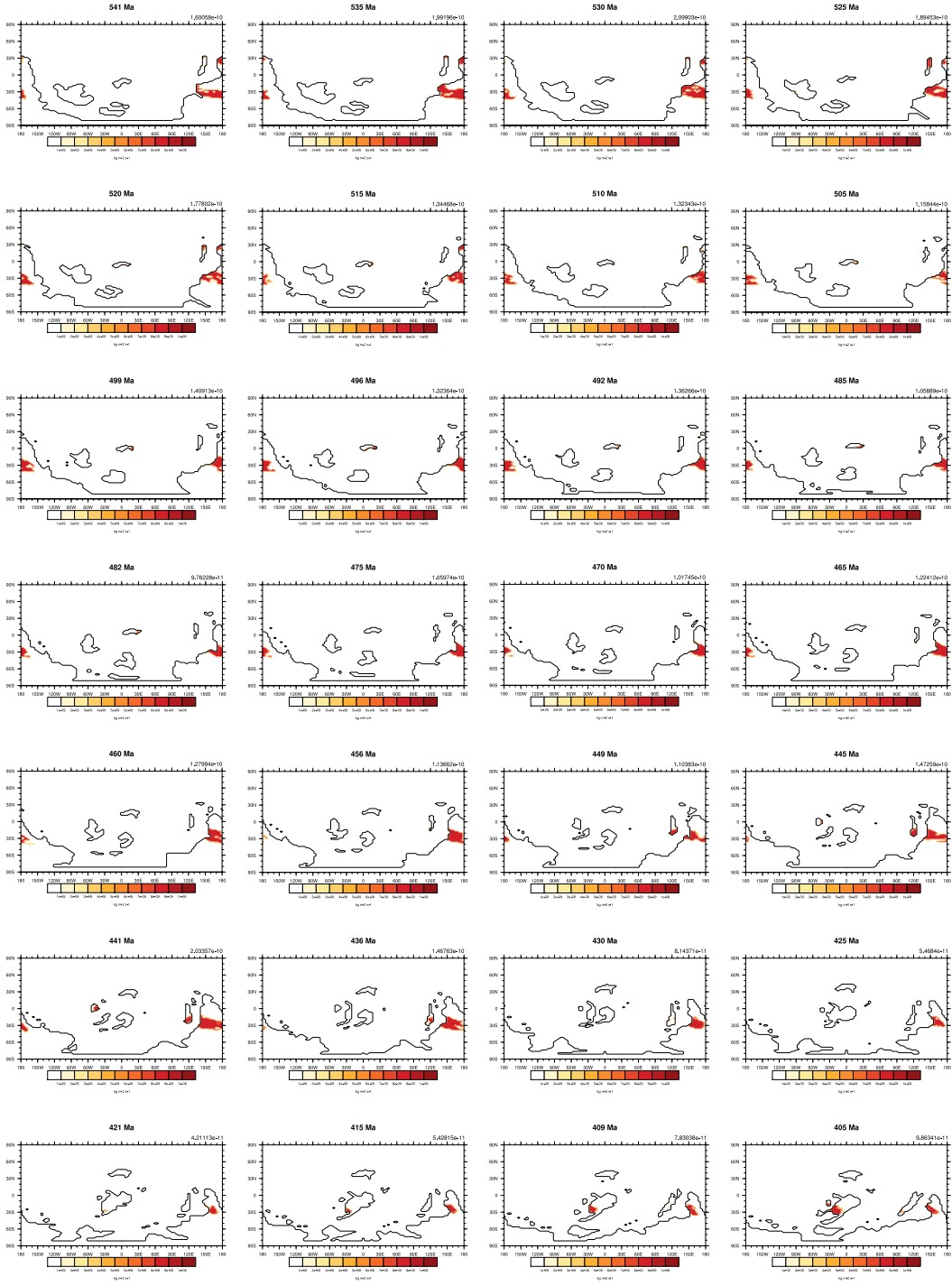

**Figure A2.** The simulated dust emissions of all 109 time slices over the Phanerozoic (Part 1). Values at the top-right are the global mean dust emission rates. Unites are $kg\ m^{-2}s^{-1}$.

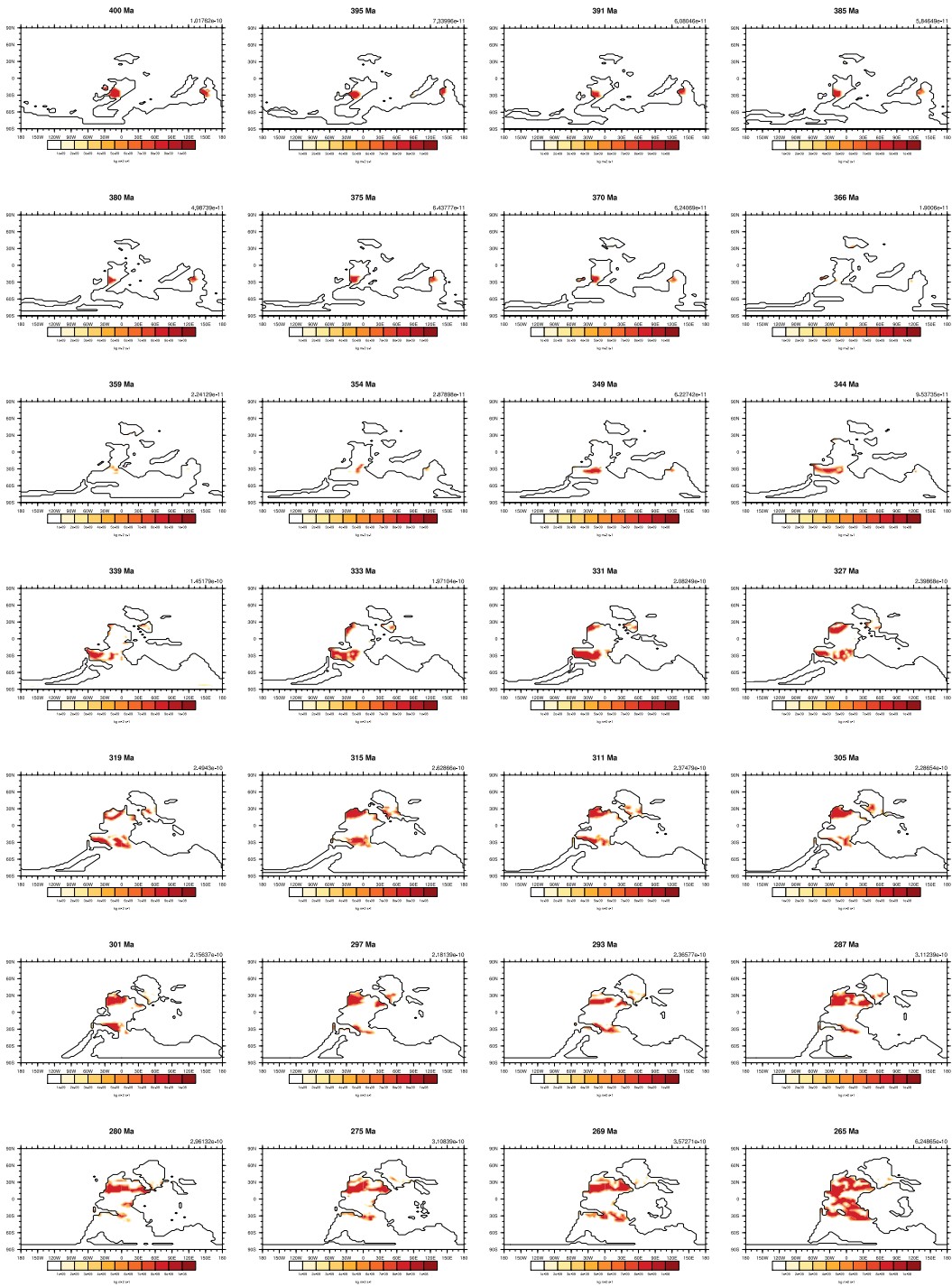

**Figure A3.** The simulated dust emissions of all 109 time slices over the Phanerozoic (Part 2). Values at the top-right are the global mean dust emission rates. Unites are $kg\ m^{-2}s^{-1}$.

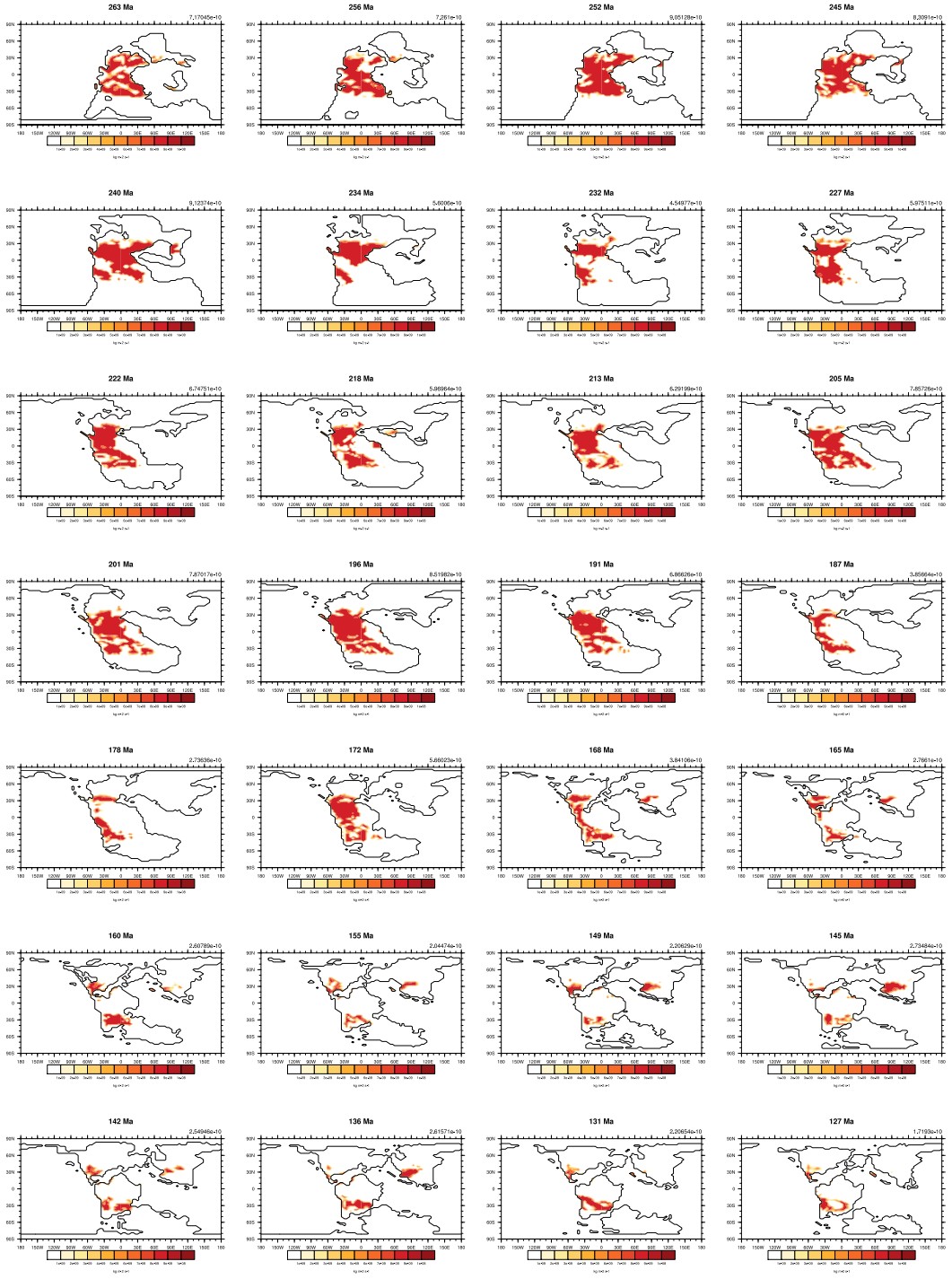

**Figure A4.** The simulated dust emissions of all 109 time slices over the Phanerozoic (Part 3). Values at the top-right are the global mean dust emission rates. Unites are $kg\ m^{-2}s^{-1}$.

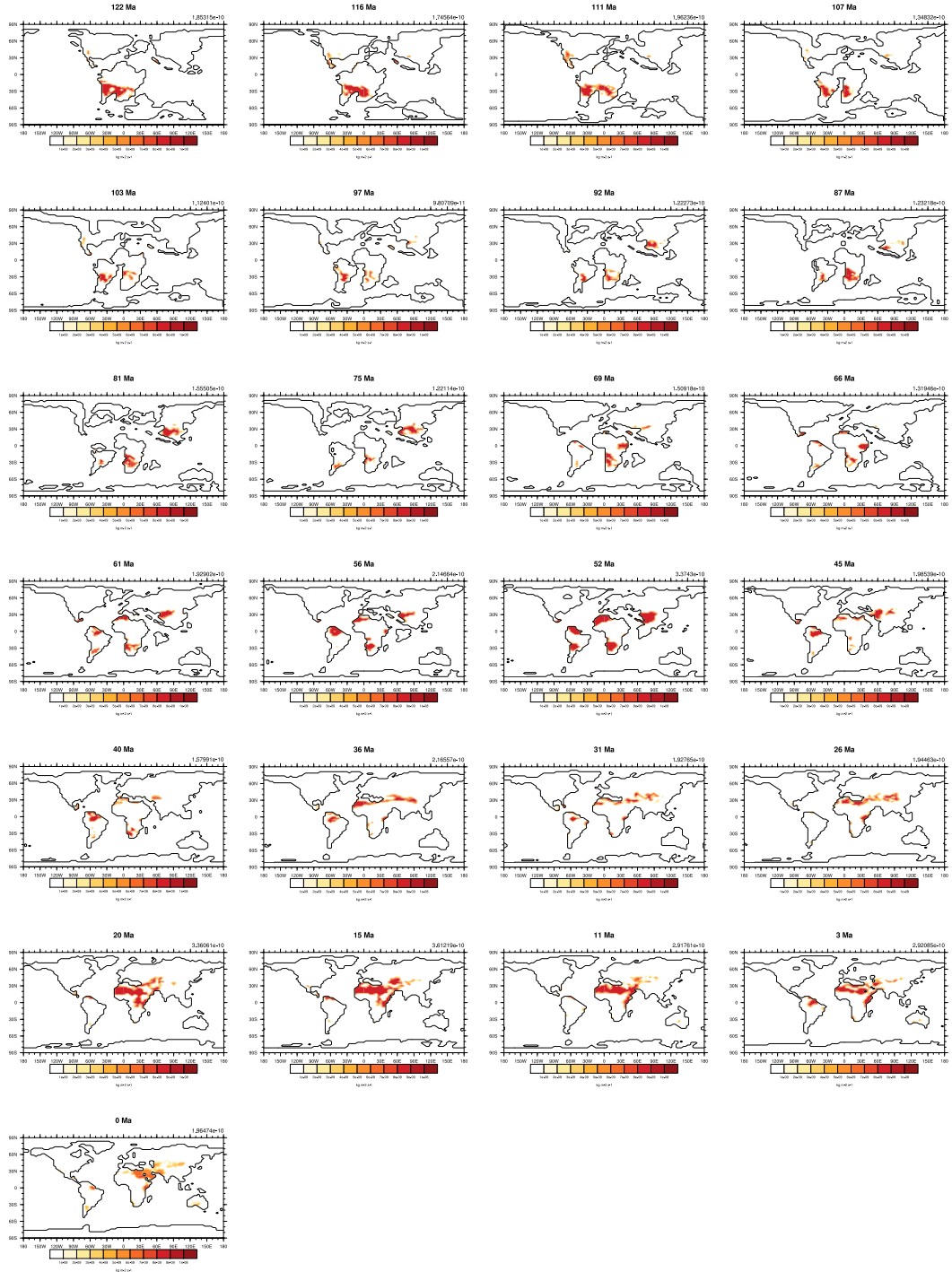

**Figure A5.** The simulated dust emissions of all 109 time slices over the Phanerozoic (Part 4). Values at the top-right are the global mean dust emission rates. Unites are $kg\ m^{-2}s^{-1}$.

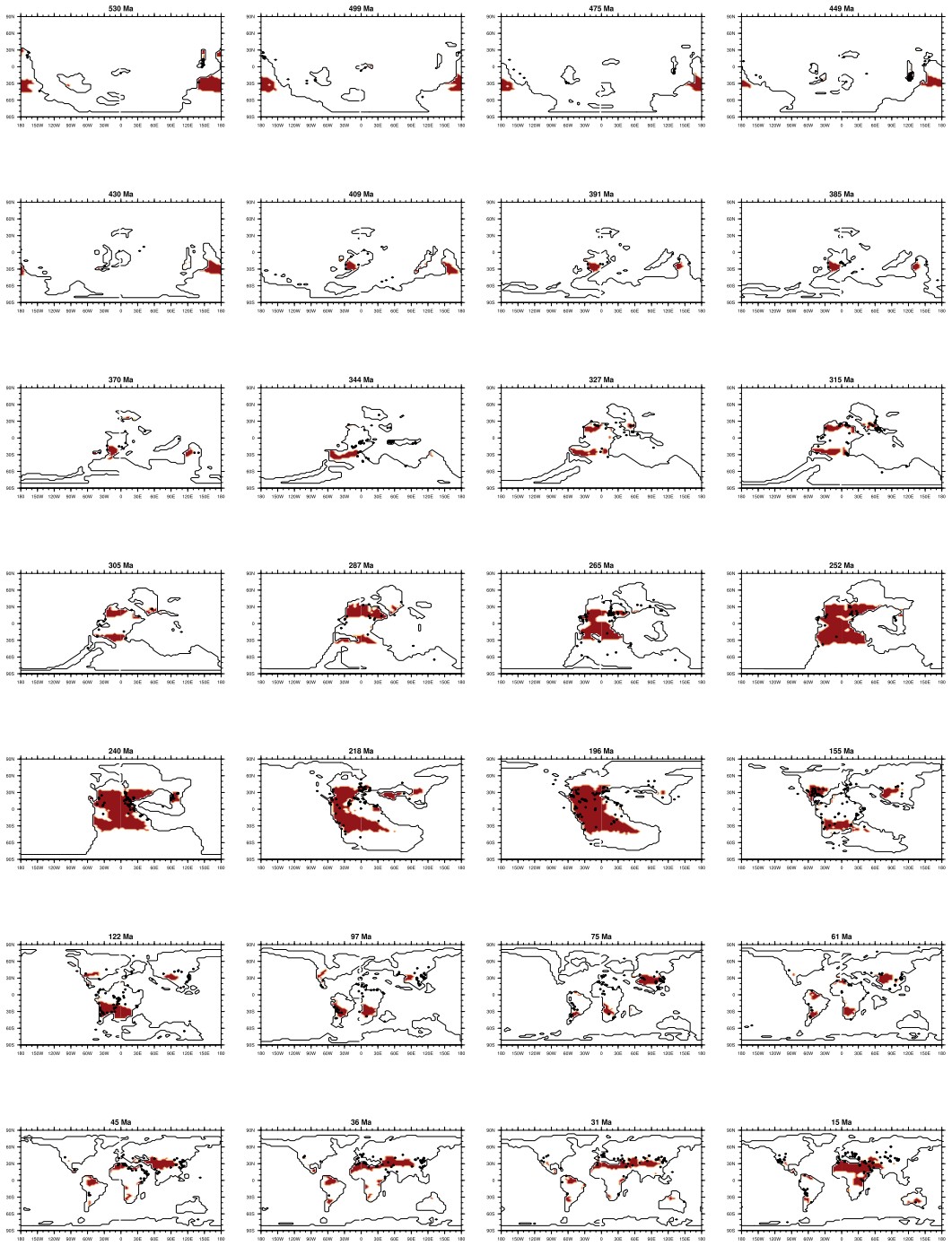

**Figure A6.** Comparisons of evaporite sedimentary records (dark dots, data from (Boucot et al., 2013)) to the simulated evaporite (area in red) of multiple time slices

*Author contributions.* The study was developed by all authors. PJV and DJL performed the original HadCM3L model simulations. DJL and YX designed the study, developed the DUSTY1.0 model and carried out the analysis.YX wrote the original manuscript, DJL and PJV provided edits .

*Competing interests.* The authors declare that they have no conflict of interest.

*Acknowledgements.* DJL and PJV acknowledge NERC grant NE/X000222/1 (PaleoGradPhan: Paleoclimate meridional and zonal Gradients in the Phanerozoic). YX acknowledges funding from the China Scholarship Council (Grant No. 202006380069).

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
