# Peer review of "Diagnosing the controls on desert dust emissions through the Phanerozoic"

_Climate of the Past, 2024_

## Author Comment (AC1)

We thank the reviewer for the constructive comments, which helped to improve our manuscript.

Below are our detailed point-by-point replies and suggested manuscript improvements (blue) for each comment (black).

In this manuscript, Xie et al. run an offline mineral dust emission model forced by a significant suite of climate model simulations that look at the broad stripes of climate change across the Phanerozoic. Based on the results of these simulations, it is argued that the principal driver of dust emission across the Phanerozoic is the degree of continentality, i.e., times when there are big continents across the subtropics. Greenhouse gas levels are of minimal import. The manuscript compares its results with the distribution of evaporite records and finds some agreement ("fair" used in the abstract may be overly generous), which is considered sufficient validation of the model. I will note that the results about the broad movements in Phanerozoic climate align well with the results of Hu et al. (2023, https://doi.org/10.1038/s41561-023-01288-y) on monsoon strength, possibly because the underlying model experiments use similar approaches.

We agree that the words have been used for describing model-data comparison is not appropriate, in the revised manuscript, we will consider using more appropriate language for summarising the model-data comparison. We will also highlight the consistency with the results of Hu et al (2023).

The work presented in this manuscript is a valuable contribution to our understanding of the geological record of dust by providing a sort of null hypothesis testable against that record: that dust emission (and the level of global deposition) are driven mainly by variability in hydroclimate (and factors directly related to hydroclimate such as vegetation cover). The experiments presented in the manuscript are suitable to provide this null hypothesis. The manuscript is clearly written and well-organised. However, the manuscript takes little account of research on the geological record of dust, either in the recent or deep past. The outcome is that evaporite distribution is used as a proxy for dust emission. However, this type of validation is circular. Evaporite deposition is, of course, more demonstrably controlled by hydroclimate and continentality. So, of course, a model that has a response strongly controlled by hydroclimate and continentality is going to match the distribution of evaporites quite well.

We agree that the geological dust record referenced in the original manuscript is not an adequate review in support of the proposition that "deep-time dust is poorly understood", in the revised manuscript, we will provide a more detailed review of the deep time geological record of dust.

We will also make it much clearer in the revised manuscript that we are not considering evaporites as a proxy for dust emissions. Instead, we are using evaporite distribution as a method for evaluating the hydrological cycle in the model. The idea of the validation is that because there are limited geological records that can directly represent dust emissions, we here carry out an indirect comparison to evaluate our simulations. The simulated evaporites

from HadCM3 are compared to the evaporite records, and this model-data comparison is intended to evaluate the performance of HadCM3 in simulating the hydrological cycle. Because our analysis shows that the dust emissions are controlled by arid regions, this is an important, albeit indirect, evaluation of the model. We do not believe that there is any circularity here.

The alternate hypothesis that these simulations should allow to be tested once further work is done to characterise the geological record of dust through time is that dust emission through time is a balance between two, sometimes competing factors. First, dust is mostly produced by physical weathering in glacial margin environments rather than "just there" or forming from the collision of desert sands. Second, dust is most easily emitted in dry, but not too dry environments, where vegetation cover is minimal but crusting together of small particles by evaporite salts is also minimal or reduced by occasional re-wetting. The upshot of this hypothesis is that dust emission increases gradually in icehouse climates as sediment availability increases, peaks in the early part of greenhouse climates (or in states of higher global continentality) following an icehouse climate, and then wanes through the course of long eras of greenhouse climate, as the dust (technically silt-sized sediment) reaches long-term sediment sinks, like the ocean. Because glacial grinding is a contested process, it is therefore very helpful to have a modelling study that removes the sediment from the equation and looking at change across the Phanerozoic. But it would be best if the manuscript gave the reader better information about the alternate hypothesis and the observables in the geological record. I therefore recommend that this manuscript be published after revision to better review the deep time geological record of dust and consider the inconsistencies the modelling approach in the manuscript might have with even the relatively recent record of dust.

We agree that weathering related to glaciation could potentially play an important role in dust emissions through its impact on surface particle availability. However, testing the importance of this factor requires higher-resolution data of glacial movement and melting, as well as topography, of the Earth's deep past, which is very limited in availability. In the revised manuscript, we will state clearly that we are not including some potentially relevant processes, including sediment availability, and summarise some of these processes. We will also highlight the uncertainties that this gives our results and highlight this as a topic for future work in the revised manuscript.

I will focus first on how the modelling in this manuscript looks in the light of geological evidence from the Holocene and Pleistocene, provide some useful background on ongoing work in reconstructing the deep time record of dust, and then provide some minor comments on the manuscript line by line. The model in this manuscript is one of atmospheric dust emission. By tuning to the AMIP models, the manuscript is strictly speaking only treating the dust sizes (0.1-10 µm) typically represented in the AMIP models on the grounds of their relevance to atmospheric radiation/clouds and presence in dust deposition records distal from emission (such as ice cores). See Mahowald et al. (2006 , https://doi.org/10.1029/2005JD006653) or Mahowald et al. (2014, https://doi.org/10.1016/j.aeolia.2013.09.002) for further discussion of these points.

We will make it clearer in the revised manuscript that the total dust emissions are representing the emission within a fixed size fraction, equivalent to the AMIP model size range to which the model is tuned.

However, the best proxies for the geographic distribution of dust emission are proximal deposits of dust, where particle sizes are bigger (and much of the mass is concentrated geologically). These are known as loesses and can be 10s or sometimes 100s of m thick. Loesses typically have particle sizes ~50 μm. Thus, if the approximate atmospheric dry deposition lifetime of 5-10 μm is 1.7 days (Mahowald et al., 2006), the lifetime of loess particles is ~ 1 hour. Remobilisation complicates the picture, and the nature of sedimentary geology is for sinks to be few at any one time. But the upshot is that the geological record of dust emission is strongly concentrated within the bounds of wherever the wind can blow dust in a few hours (~ 100 km). So when faced with a figure showing the world's loess deposition in recent sediments, any model that connects dust emission to hydroclimate etc. rather than sediment availability is faced with a problem. (See Fig. 2 of https://doi.org/10.1016/j.earscirev.2019.102947) for a good example. Loess is abundant in mid-latitude and some high-latitude areas across North America and Eurasia, South America and parts of Australasia that include New Zealand. And it is relative rare along the southern margin of the Sahara. The model presented in the manuscript might explain the rarity of Saharan loess for the most recent time slice. (But where are the Pliocene loesses in the Sahara contemporary with the Eurasian ones?) But North American loess, European loess, and New Zealand loesses are inexplicable. This observation is the motivation for the idea that dust emission is shaped by sediment availability, with sediment primarily being generated by physical weathering at glacial margins (see discussion in Mahowald et al., 2006 and Prospero et al., 2002, https://doi.org/10.1016/j.earscirev.2019.102947).

The original motivation for this work was to explore the climatic impact of changing dust emissions through time, in particular in terms of changes in marine biogeochemistry and iron fertilisation. The ultimate aim was to produce "maps" of dust deposition over oceans that could ultimately be input into marine biogeochemical models. Hence our focus on the smaller longer-lived particle sizes. We agree that this means that we are not simulating larger particles, which are more evident in the terrestrial geological record. In the revised manuscript we will explain the reasons for this focus, and also the implications of this in terms of model-data comparisons. In particular, we will highlight that by not explicitly including dust sources associated with glacial processes, we are unlikely to correctly simulate high latitude dust sources.

The manuscript certainly demonstrates limited knowledge of dust records through geological time. At line 49, it says, "similar orbital-driven dust (or aridity) records are also identified much further back in time, as early as the Late Cretaceous 50 (Niedermeyer et al., 2010; Vallé et al., 2017; Zhang et al., 2019)." In fact, the most widespread evidence for orbitally-driven dust deposition is from the Late Paleozoic. A good place to start would be a recent review by Soreghan et al. (https://doi.org/10.1144/SP535-2022-208) and probably too recent for the authors to see (https://doi.org/10.2110/jsr.2023.122). The work by Lynn Soreghan suggests loess deposition was widespread, even at low latitudes in late Carboniferous and early Permian Pangaea. Loess also seems widespread in much of the Permian (where we can see), not just in places Soreghan and her collaborators have looked but in Russia (Mouraviev et al.,

2015, doi: https://doi.org/10.1134/S0031030115110064). Loess deposition certainly continues deeper into the long greenhouse interval (see Chan et al., 1999, https://doi.org/10.2110/jsr.69.477), but the Triassic peak in dust emission by the model seems inconsistent with the record, where the places we would expect to see loesses are mostly sandstones and evaporites.

Many thanks to the reviewer for providing useful reference on the paleo dust records, we will make use of these suggested reference in providing an improved review of dust records and comparison with our simulations.

For the inconsistency mentioned by the reviewer regarding the Triassic peak: The location (approximately 54°W, 17°N, rotated according to the GPlate model for the upper Triassic continent) where loess sediment was identified by Chan et al., 1999, is simulated to be an area where desert dust and evaporites could form (see plots for 218 Ma in both Figure A4 and Figure A6). We do not believe this indicates an inconsistency between our simulation and geological records. Firstly, the spatial resolution of our simulations is significantly different from the actual length of the cross-section where the loess is found. Therefore, for a grid box where the model predicts desert/evaporite, we cannot assume that the entire land area of that grid box is covered by desert/evaporite. Similarly, we cannot deny the possibility that loess may exist within that grid area. Secondly, both the sediment record and model simulation times are approximations, both around 213 Ma, with considerable uncertainty regarding the exact age. Therefore, we cannot conclude there is an inconsistency based on the fact that the absolute ages of the sediment record and model simulation do not precisely align.

I still do agree with the general thrust of the manuscript that the deep time record of dust is poorly understood, but this should not be done without emphasising the importance of loess as a proxy for dust emission, some more expanded references to work in this area, and more nuance when comparing the model results with evaporite distributions.

**Minor Comments**

Line 47: This is probably where sediment availability and glacial grinding should appear as a hypothesis. Additional possible references are: Reader et al. 1999 (https://doi.org/10.1029/1999JD900033) and Sugden et al. (2009, https://doi.org/10.1038/ngeo474)

Yes, we will add words about sediment availability and reference in the corresponding paragraph.

Line 100: Omitting the Zender et al. formulation is justifiable if you are clear that you are trying to help test between hydroclimate/meteorology and sediment supply hypotheses.

We'll make it clearer in the revised manuscript that our model simulations cannot test the sediment supply hypotheses .

Figure 1: The notation of units in the legend/colorbar is confusing. Normally, you don't superscript the exponential notation, so there's a clear distinction between e-09=10^-9 vs. e^-9

We'll correct and uniform the units in figures.

Line 339: Desert has become dessert

Corrected.

Fig. A4 Part 3: 213 Ma looks like a plotting error. Please check.

It was a wrong plot. We'll update it in the revised manuscript.

---

## Author Comment (AC2)

Reply to Referee #2 Yonggang Liu

We appreciate the constructive and detailed comments from the reviewer. Below we reply to each point they raise (black) and explain how we intend to revise the manuscript (blue).

Xie et al. develop and tune a new offline dust emission model, DUSTY, based on which and the climate simulations they have carried out previously, a continuous model-derived timeseries of global dust emissions over the whole Phanerozoic is established. The simulation results provide quantitative insights into significant fluctuations in dust emissions during this Era. Notably, the study highlights the dominant influence of non-vegetated areas on dust emissions. The authors further investigate the mechanisms underlying the hydrological variations, governs the aridity the distribution non-vegetated areas. The findings demonstrate that paleogeography serves as the primary driving force behind dust emission variations, while the role of $CO_2$ is found to be marginal. The comparison between the simulated region of dust emission and the distribution of evaporite sediment records is not bad given the uncertainties in so many boundary conditions and model parameters. The manuscript provides new knowledge to both geologists and paleoclimatologists and clearly written in general, I recommend publication after a relatively minor revision.

**Major comments**

1. More detailed comparison should be made here of the 15_MMM and 0Ma simulations, which are crucial for tuning and evaluating the model. Although the model parameters of the DUSTY model have been optimized by maximizing the Arcsin Mielke score and the global mean emissions are the same, significant differences still exist between Figures 1a and 1b. For example, the dust emission in the middle of Eurasia and South America is seriously underestimated while that over Australia is highly overestimated. Are they due to biases in the simulated climate, vegetation or dust parameterization? The implication to the simulated distribution of dust emission in other periods should also be discussed.

   The comparison between the tuning target (Figure 1a) and the tuned pre-industrial simulation (Figure 1b) indicates that the DUSTY model can represent the general spatial distribution of dust emissions. However, biases exist in the detailed distributions. In Figure 1a, dust emissions have hotspots in regions such as the eastern Sahara and central Asia. In contrast, Figure 1b shows emissions more evenly distributed, lacking standout areas with high emissions. This discrepancy is largely due to the absence of detailed morphology and particle size representation in our models. These factors are crucial for capturing small-scale areas with frequent dust activation, such as the eastern Sahara. Because we force the DUSTY model to match the global mean emission rate of the target, the average emission in each sub-area is enhanced in the absence of hotspots. This leads to overestimation in regions like Australia and western Sahara. While vegetation should not be attributed to the bias in this comparison because, in the standard pre-industrial simulations the vegetation is from observations instead of simulated from TRIFFID (described in line 166 -169 in the original manuscript).

Further, in terms of the simulated dust field in paleo time slices, our model is expected not able to simulate small-scale areas with standout emissions, and will overestimate the emission rate over the broader areas as well. Additionally, the paleo simulations involve greater uncertainty in vegetation and topography compared to the standard pre-industrial run, which introduces additional biases into the simulated dust emission fields.

We will add the above discussion to the revised manuscript.

2. The absence of land plant colonisation before 410Ma implies that the inclusion of vegetation cover during 541-410Ma in this study may have resulted in an underestimation of dust emission during this period. The influence of this effect on the major conclusions of their study should also be discussed.

The TRIFFID vegetation scheme, coupled with the GCM, employs a simply representation of terrestrial vegetation, incorporating five plant functional types that cover a broad climate range. These plant types do not go extinct and can regenerate under suitable conditions. This simplification introduces a bias for early Paleozoic periods before the colonization of land plants, resulting in our simulations predicting both bare soil and vegetation where, in reality, there would have been only bare soil. Consequently, this leads to an underestimation of dust emissions during those period.

This issue explains the poor consistency with evaporite records in the time slices before the Devonian (Figure 9 and Figure A6), along with biases potentially introduced by GCM simulation and DUSTY parameterization. A more quantitative evaluation of the sensitivity of dust emissions to dynamic vegetation can be achieved through parallel simulations excluding the TRIFFID model. Sensitivity tests in Liu et al. (2020) demonstrate that including dynamic vegetation can lead to up to a fourfold underestimation of dust emissions in the Precambrian, highlighting the significant impact of non-existent land vegetation.

We will add this discussion to the limitations in Section 5.2 and address the higher uncertainty before land plant colonization in Section 4.1, describing the paleo dust emission results, as well as in the Conclusion section in the revised manuscript.

**Specific Comments**

**Line 45-48:** May add "glaciogenic rock powder" or other equivalent expressions.

Yes, will add "more abundant sediment availability" and reference in the corresponding paragraph.

**Line 188-190:** is the global mean the average over all the land or over the regions with nonzero dust emission?

The global mean is calculated over the entire global area, including both land and ocean. This approach ensures consistency when comparing the means across different time slices and avoids confusion arising from variations in the total land area over time.

**Line 189:** 'C1' should be changed to 'C2'. This correction is also required in Table 1.

Thanks for pointing this out! They will be corrected in the revised manuscript.

**Line 230:** It should be added here that the results are from the S2 series of experiments.

Sure, we will clarify the experiment series name at the beginning of introducing paleo results in the revised manuscript.

**Line 282-285:** How was the value of 0.8 mm/day chosen here? The bracket at the end of Equation 10 needs to be removed.

The value of 0.8 mm/day is chosen based on studies that have shown that sediments, such as evaporites and weathered sandstones, form under precipitation conditions below this threshold, indicating an arid environment. This is an approximation referenced from Watson et al., 1992; Price et al., 1994; Cecil et al., 2003; and Craggs et al., 2012. We will add a justification for this value in the revised manuscript.

And the extra bracket will be removed.

References:

Cecil, C. B.: The Concept of Autocyclic and Allocyclic Controls on Sedimentation and Stratigraphy, Emphasizing the Climatic Variable, https://doi.org/10.2110/pec.03.77.0013, 2003.

Craggs, H. J., Valdes, P. J., and Widdowson, M.: Climate model predictions for the latest Cretaceous: An evaluation using climatically sensitive sediments as proxy indicators, Palaeogeography, Palaeoclimatology, Palaeoecology, 315-316, 12–23, https://doi.org/10.1016/j.palaeo.2011.11.004, 2012.

Liu, P., Liu, Y., Peng, Y., Lamarque, J. F., Wang, M., and Hu, Y.: Large influence of dust on the Precambrian climate, Nature Communications, 11, 1–8, https://doi.org/10.1038/s41467-020-18258-2, 2020.

Warren, J. K.: Evaporites through time: Tectonic, climatic and eustatic controls in marine and nonmarine deposits, Earth-Science Reviews,98, 217–268, https://doi.org/10.1016/j.earscirev.2009.11.004, 2010.

---

## Author Response (AR1)

We thank the reviewers for their careful and detailed comments, which have greatly improved the manuscript. Below is our response to these comments; reviewers' comments are in normal font; **our responses are in bold font.** Line numbers in blue refer to the "latexdiff" comparison of the revised manuscript with the original manuscript, appended at the end of this response.

Reply to the Anonymous Referee #1

In this manuscript, Xie et al. run an offline mineral dust emission model forced by a significant suite of climate model simulations that look at the broad stripes of climate change across the Phanerozoic. Based on the results of these simulations, it is argued that the principal driver of dust emission across the Phanerozoic is the degree of continentality, i.e., times when there are big continents across the subtropics. Greenhouse gas levels are of minimal import. The manuscript compares its results with the distribution of evaporite records and finds some agreement ("fair" used in the abstract may be overly generous), which is considered sufficient validation of the model. I will note that the results about the broad movements in Phanerozoic climate align well with the results of Hu et al. (2023, https://doi.org/10.1038/s41561-023-01288-y) on monsoon strength, possibly because the underlying model experiments use similar approaches.

**We have changed the wording from "fair agreement" to "reasonable agreement" in the abstract and the conclusion in the revised manuscript. Line 16, 498.**

The work presented in this manuscript is a valuable contribution to our understanding of the geological record of dust by providing a sort of null hypothesis testable against that record: that dust emission (and the level of global deposition) are driven mainly by variability in hydroclimate (and factors directly related to hydroclimate such as vegetation cover). The experiments presented in the manuscript are suitable to provide this null hypothesis. The manuscript is clearly written and well-organised. However, the manuscript takes little account of research on the geological record of dust, either in the recent or deep past. The outcome is that evaporite distribution is used as a proxy for dust emission. However, this type of validation is circular. Evaporite deposition is, of course, more demonstrably controlled by hydroclimate and continentality. So, of course, a model that has a response strongly controlled by hydroclimate and continentality is going to match the distribution of evaporites quite well.

**We clarified in section 5.1 revised manuscript that we are not considering evaporites as a proxy for dust emissions. Instead, we are using evaporite distribution as a method for evaluating the hydrological cycle in the model. The idea of the validation is that because there are limited geological records that can directly represent dust emissions, we instead here carry out an indirect comparison to evaluate our climate model simulations which drive the dust model. The simulated evaporites from HadCM3 are compared to the evaporite records, and this model-data comparison is intended to evaluate the performance of HadCM3 in simulating the hydrological cycle. Because our analysis shows that the dust emissions are controlled by arid regions, this is an important, albeit indirect, evaluation of the modelling framework. We do not believe that there is any circularity here. Lines 375-388.**

The alternate hypothesis that these simulations should allow to be tested once further work is done to characterise the geological record of dust through time is that dust emission through time is a balance between two, sometimes competing factors. First, dust is mostly produced by physical weathering in glacial margin environments rather than "just there" or forming from the collision of desert sands. Second, dust is most easily emitted in dry, but not too dry environments, where vegetation cover is minimal but crusting together of small particles by evaporite salts is also minimal or reduced by occasional re-wetting. The upshot of this hypothesis is that dust emission increases gradually in icehouse climates as sediment availability increases, peaks in the early part of greenhouse climates (or in states of higher global continentality) following an icehouse climate, and then wanes through the course of long eras of greenhouse climate, as the dust (technically silt-sized sediment) reaches long-term sediment sinks, like the ocean. Because glacial grinding is a contested process, it is therefore very helpful to have a modelling study that removes the sediment from the equation and looking at change across the Phanerozoic. But it would be best if the manuscript gave the reader better information about the alternate hypothesis and the observables in the geological record. I therefore recommend that this manuscript be published after revision to better review the deep time geological record of dust and consider the inconsistencies the modelling approach in the manuscript might have with even the relatively recent record of dust.

**We agree that weathering related to glaciation could potentially play an important role in dust emissions through its impact on surface particle availability. However, testing the importance of this factor requires higher-resolution data of glacial movement and melting, as well as topography, of the Earth's deep past, which is very limited in availability. In the revised manuscript, we added literature that highlights this important mechanism, both from modern observations and paleo studies, and clarified our model cannot directly test this hypothesis. We also suggested future research based on this limitation. Lines 49-51, 58, 107-111, 486-488.**

I will focus first on how the modelling in this manuscript looks in the light of geological evidence from the Holocene and Pleistocene, provide some useful background on ongoing work in reconstructing the deep time record of dust, and then provide some minor comments on the manuscript line by line. The model in this manuscript is one of atmospheric dust emission. By tuning to the AMIP models, the manuscript is strictly speaking only treating the dust sizes (0.1-10 μm) typically represented in the AMIP models on the grounds of their relevance to atmospheric radiation/clouds and presence in dust deposition records distal from emission (such as ice cores). See Mahowald et al. (2006 , https://doi.org/10.1029/2005JD006653) or Mahowald et al. (2014, https://doi.org/10.1016/j.aeolia.2013.09.002) for further discussion of these points.

**We clarified the particle size our model takes into consideration by referring to the parameterization of the AMIP models that we are turning to. Lines 206-208.**

However, the best proxies for the geographic distribution of dust emission are proximal deposits of dust, where particle sizes are bigger (and much of the mass is concentrated

geologically). These are known as loesses and can be 10s or sometimes 100s of m thick. Loesses typically have particle sizes ~50 μm. Thus, if the approximate atmospheric dry deposition lifetime of 5-10 μm is 1.7 days (Mahowald et al., 2006), the lifetime of loess particles is ~ 1 hour. Remobilisation complicates the picture, and the nature of sedimentary geology is for sinks to be few at any one time. But the upshot is that the geological record of dust emission is strongly concentrated within the bounds of wherever the wind can blow dust in a few hours (~ 100 km). So when faced with a figure showing the world's loess deposition in recent sediments, any model that connects dust emission to hydroclimate etc. rather than sediment availability is faced with a problem. (See Fig. 2 of https://doi.org/10.1016/j.earscirev.2019.102947) for a good example. Loess is abundant in mid-latitude and some high-latitude areas across North America and Eurasia, South America and parts of Australasia that include New Zealand. And it is relative rare along the southern margin of the Sahara. The model presented in the manuscript might explain the rarity of Saharan loess for the most recent time slice. (But where are the Pliocene loesses in the Sahara contemporary with the Eurasian ones?) But North American loess, European loess, and New Zealand loesses are inexplicable. This observation is the motivation for the idea that dust emission is shaped by sediment availability, with sediment primarily being generated by physical weathering at glacial margins (see discussion in Mahowald et al., 2006 and Prospero et al., 2002, https://doi.org/10.1016/j.earscirev.2019.102947).

The manuscript certainly demonstrates limited knowledge of dust records through geological time. At line 49, it says, "similar orbital-driven dust (or aridity) records are also identified much further back in time, as early as the Late Cretaceous 50 (Niedermeyer et al., 2010; Vallé et al., 2017; Zhang et al., 2019)." In fact, the most widespread evidence for orbitally-driven dust deposition is from the Late Paleozoic. A good place to start would be a recent review by Soreghan et al. (https://doi.org/10.1144/SP535-2022-208) and probably too recent for the authors to see (https://doi.org/10.2110/jsr.2023.122). The work by Lynn Soreghan suggests loess deposition was widespread, even at low latitudes in late Carboniferous and early Permian Pangaea. Loess also seems widespread in much of the Permian (where we can see), not just in places Soreghan and her collaborators have looked but in Russia (Mouraviev et al., 2015, doi: https://doi.org/10.1134/S0031030115110064). Loess deposition certainly continues deeper into the long greenhouse interval (see Chan et al., 1999, https://doi.org/10.2110/jsr.69.477), but the Triassic peak in dust emission by the model seems inconsistent with the record, where the places we would expect to see loesses are mostly sandstones and evaporites.

**Many thanks to the reviewer for providing useful references on the paleo dust records; we made use of this suggested reference and improved the review of dust records. Lines 49-51, 58, 108.**

**For the inconsistency mentioned by the reviewer regarding the Triassic peak: The location (approximately 54°W, 17°N, rotated according to the GPlate model for the upper Triassic continent) where loess sediment was identified by Chan et al., 1999, is simulated to be an area where evaporites could form (see plot for 218 Ma in Figure A6). We do not believe this indicates an inconsistency between our simulation and geological records. Firstly, the spatial resolution of our simulations is significantly different from the actual length of the cross-section where the loess is found. Therefore, for a grid box where the model predicts evaporite, we cannot assume that the entire land area of that grid box is covered by evaporite. Similarly, we cannot deny the possibility that loess may exist within that grid area. Secondly, both the sediment**

**record and model simulation times are approximations, both around 218 Ma, with considerable uncertainty regarding the exact age. Therefore, we cannot conclude there is an inconsistency based on the fact that the absolute ages of the sediment record and model simulation do not precisely align.**

I still do agree with the general thrust of the manuscript that the deep time record of dust is poorly understood, but this should not be done without emphasising the importance of loess as a proxy for dust emission, some more expanded references to work in this area, and more nuance when comparing the model results with evaporite distributions.

Minor Comments

Line 47: This is probably where sediment availability and glacial grinding should appear as a hypothesis. Additional possible references are: Reader et al. 1999 (https://doi.org/10.1029/1999JD900033) and Sugden et al. (2009, https://doi.org/10.1038/ngeo474)

**We added words about sediment availability and reference in the corresponding paragraph. Lines 49-51, 58.**

Line 100: Omitting the Zender et al. formulation is justifiable if you are clear that you are trying to help test between hydroclimate/meteorology and sediment supply hypotheses.

**We have made it clearer in the revised manuscript that our model simulations cannot test the sediment supply hypotheses. Lines 108-111, 486-488.**

Figure 1: The notation of units in the legend/colorbar is confusing. Normally, you don't superscript the exponential notation, so there's a clear distinction between e-09=10^-9 vs. e^-9

**We have corrected the notation of units in figures.**

Line 339: Desert has become dessert

**Corrected.**

Fig. A4 Part 3: 213 Ma looks like a plotting error. Please check.

**We have corrected this error in the revised manuscript.**

Xie et al. develop and tune a new offline dust emission model, DUSTY, based on which and the climate simulations they have carried out previously, a continuous model-derived timeseries of global dust emissions over the whole Phanerozoic is established. The simulation results provide quantitative insights into significant fluctuations in dust emissions during this Era. Notably, the study highlights the dominant influence of non-vegetated areas on dust emissions. The authors further investigate the mechanisms underlying the hydrological variations, governs the aridity the distribution non-vegetated areas. The findings demonstrate that paleogeography serves as the primary driving force behind dust emission variations, while the role of $CO_2$ is found to be marginal. The comparison between the simulated region of dust emission and the distribution of evaporite sediment records is not bad given the uncertainties in so many boundary conditions and model parameters. The manuscript provides new knowledge to both geologists and paleoclimatologists and clearly written in general, I recommend publication after a relatively minor revision.

Major comments

1. More detailed comparison should be made here of the 15_MMM and 0Ma simulations, which are crucial for tuning and evaluating the model. Although the model parameters of the DUSTY model have been optimized by maximizing the Arcsin Mielke score and the global mean emissions are the same, significant differences still exist between Figures 1a and 1b. For example, the dust emission in the middle of Eurasia and South America is seriously underestimated while that over Australia is highly overestimated. Are they due to biases in the simulated climate, vegetation or dust parameterization? The implication to the simulated distribution of dust emission in other periods should also be discussed.

   **The comparison between the tuning target (Figure 1a) and the tuned pre-industrial simulation (Figure 1b) indicates that the DUSTY model can represent the general spatial distribution of dust emissions. However, biases exist in the detailed distributions. It represents our simulations fail to capture the small-scale areas with very high emissions (hot spots). We added contents in section 3.2. And expand the similar potential discrepancy in section 4.1. Lines 248-254, 275-277.**

2. The absence of land plant colonisation before 410Ma implies that the inclusion of vegetation cover during 541-410Ma in this study may have resulted in an underestimation of dust emission during this period. The influence of this effect on the major conclusions of their study should also be discussed.

   **The TRIFFID vegetation scheme, coupled with the GCM, employs a simple representation of terrestrial vegetation, incorporating five plant functional types that cover a broad climate range. This simplification introduces a bias for early Paleozoic periods before the colonization of land plants, resulting in our simulations predicting both bare soil and vegetation where, in reality, there would**

**have been only bare soil. Consequently, this leads to an underestimation of dust emissions during those periods.**

**We carried out further simulations to constrain the underlaying uncertain and added this discussion to the limitations in Section 5.2. The underestimation is also added to the conclusion. Lines 447-462, 497.**

 Specific Comments

Line 45-48: May add "glaciogenic rock powder" or other equivalent expressions.

**We added contents about sediment availability. Also clarified in the revised manuscript that our model simulations cannot test the sediment supply hypotheses. Lines 49-51, 58, 107-110**

Line 188-190: is the global mean the average over all the land or over the regions with nonzero dust emission?

**The global mean is calculated over the entire global area, including both land and ocean. This approach ensures consistency when comparing the means across different time slices and avoids confusion arising from variations in the total land area over time.**

Line 189: 'C1' should be changed to 'C2'. This correction is also required in Table 1.

**Both are corrected.**

Line 230: It should be added here that the results are from the S2 series of experiments.

**We added clarification. Line 260.**

Line 282-285: How was the value of 0.8 mm/day chosen here? The bracket at the end of Equation 10 needs to be removed.

**We added reference on choosing this value. And the extra bracket has been removed. Lines 318-319.**

Reply to the Editor Yannick Donnadieu

Technical points to start:
• You say that the dust model requires high-frequency outputs, but it remains difficult to understand how these outputs are used to calculate the different parameters and why it is not possible to do so with monthly outputs.

**The GCM outputs used to run the dust model in this study are of hourly resolution, rather than monthly. This is because dust emissions have a cubic dependence on windspeed, and so it is important to capture short timescale variations in windspeed in order to correctly simulate dust emissions.**

• In addition, how are the C constants estimated? Are they calculated point by point or is it a global average value?

**We clarified the coefficients in section 2.1. They are all calculated at a global mean level. Lines 126-129.**

• Is there a $CO_2$ fertilization effect on continental vegetation? And if so, what is the effect on the prediction of arid areas for past times with high $CO_2$ concentrations?

**Yes, the vegetation model TRIFFID sees the varying $CO_2$. Thus, we should see fewer non-vegetation areas for past times with high $CO_2$ concentrations. It would be possible to explore this further, with a sensitivity study in which the TRIFFID model saw a constant $CO_2$, but this is beyond the scope of this manuscript. We added discussion on this aspect of uncertainty in section 4.2. Lines 469-475.**

• For climate simulations with constant CO2, does the solar constant vary?

**Yes, the solar constants used to drive the simulation series with constant CO2 varies the same as other simulation series. We have added sentences in section 2.3 to clarify. Lines 176-177.**

• Figure 6 is cited in the text after Figures 7 and 8, please change the numbering.

**Changed.**

Two moderate points:
• Why not directly analyze the outputs of the vegetation model, in particular the distribution of arid areas? Why calculate an aridity index when the climate model can give you direct access to arid areas? In this logic, would it be possible to directly plot the evolution of arid areas from the vegetation model? The last sentences of the conclusion suggest that the evolution of dust flux would only be linked to the continents that cross the subtropical arid

zones, it is not that simple and geometric. Figure 8 shows very well that the major increase in dust flux is also a consequence of the collapse of the tropical humid zone between 280 and 190 Ma. Atmospheric and oceanic circulation diagnostics are missing to provide an explanation for the reasons for the disappearance of the tropical humid zone.

**We do see similar patterns of the non-vegetated-area evolution to the evolution of arid areas. The primary causal relationship is likely that the aridity leads to lack of vegetation, but there is also of course dynamic feedback from the vegetation to the climate. As such, showing the aridity index as a way of explaining the non-vegetated areas is appropriate.**

**We added interpretations to explain the collapse of tropical humid zone that seen during the supercontinent times, and this is largely due to the albedo related feedback. Lines 165-169, 354-358.**

• Can you explicitly write the numerical method used to predict evaporites from your model? From my understanding of evaporite formations, they require a part of aridity but also water, therefore rather a marked seasonality with a wet season. In addition, there are also evaporites linked to marine incursions.

**We added description of the evaporite prediction scheme used in the model and explained the rationality of the comparison in section 5.1. Evaporites linked to marine incursions are excluded by kicking off those 
[revised manuscript text omitted]